# Implicit Transfer Operator Learning: Multiple Time-Resolution Surrogates for Molecular Dynamics

**Mathias Schreiner**[*]
DTU[†],
Chalmers University of Technology
matschreiner@gmail.com

**Ole Winther**
DTU

**Simon Olsson**[‡]
Chalmers University of Technology
simonols@chalmers.se

## Abstract

Computing properties of molecular systems rely on estimating expectations of the (unnormalized) Boltzmann distribution. Molecular dynamics (MD) is a broadly adopted technique to approximate such quantities. However, stable simulations rely on very small integration time-steps ($10^{-15}$ s), whereas convergence of some moments, e.g. binding free energy or rates, might rely on sampling processes on time-scales as long as $10^{-1}$ s, and these simulations must be repeated for every molecular system independently. Here, we present Implicit Transfer Operator (ITO) Learning, a framework to learn surrogates of the simulation process with multiple time-resolutions. We implement ITO with denoising diffusion probabilistic models with a new SE(3) equivariant architecture and show the resulting models can generate self-consistent stochastic dynamics across multiple time-scales, even when the system is only partially observed. Finally, we present a coarse-grained CG-SE3-ITO model which can quantitatively model all-atom molecular dynamics using only coarse molecular representations. As such, ITO provides an important step towards multiple time- and space-resolution acceleration of MD. Code is available at https://github.com/olsson-group/ito.

## 1 Introduction

Numerical simulation of stochastic differential equations (SDE) is critical in the sciences, including statistics, physics, chemistry, and biology applications [1]. Molecular dynamics (MD) simulations are an important example of such simulations [2]. These simulations prescribe a set of mechanistic rules governing the time evolution of a molecular system through numerical integration of, for example, the Langevin equation [3]. MD grants mechanistic insights into experimental observables. These observables are expectations, including time-correlations, of observable functions (e.g., pairwise distances or angles) computed for the Boltzmann distribution $\hat{\mu}(\mathbf{x}) \propto \exp[-\beta U(\mathbf{x})]$ corresponding to the *potential* $U(\cdot) : \Omega \rightarrow \mathbb{R}$ of a $M$-particle molecular system, $\mathbf{x} \in \Omega \subset \mathbb{R}^{3M}$ kept at the inverse temperature $\beta = 1/kT$. However, stable numerical integration relies on time steps, $\tau$, which are strictly smaller than the fastest characteristic time-

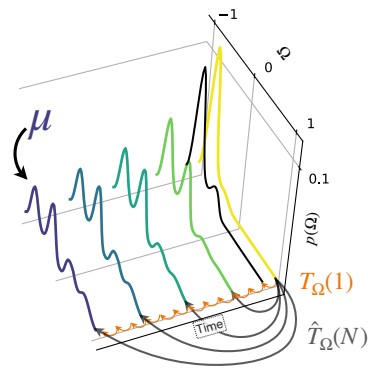

Figure 1: **Implicit Transfer Operator:** A multiple time-scale surrogate of stochastic molecular dynamics.

---

[*]Contributions to this work were done while visiting Chalmers University of Technology

[†]Technical University of Denmark

[‡]Corresponding author

37th Conference on Neural Information Processing Systems (NeurIPS 2023).

scales of the molecular system ($10^{-15}$ s, e.g., bond vibrations), yet many molecular systems are characterized by processes on much longer time-scales ($10^{-3} - 10^{-1}$ s, e.g. protein-folding, protein-ligand unbinding, regulation). Consequently, we need infeasibly long simulations to characterize many important processes quantitatively due to the slow mixing in $\Omega$.

In this work, we present the implicit Transfer Operator (ITO, Fig. 1) as an effective way to learn multiple time-step surrogate models of the stochastic generating distribution of MD. To our knowledge, this is the first surrogate modeling approach that allows for the simultaneous generation of stochastic dynamics at multiple different time resolutions. By adopting an SE(3)-equivariant generative model, we further demonstrate stable long-time-scale dynamics in increasingly difficult settings where an increasing number of degrees of freedom are marginalized. Our approach can be several orders of magnitude more efficient than direct MD simulations and can be made asymptotically unbiased if the generative model permits exact likelihood evaluation. Our current results do not generalize across different thermodynamic ensembles or across chemical space, but show strong generalization across different time-scales.

Our main contributions are

1. the **Implicit Transfer Operator (ITO)** framework for learning generative models for multiple time resolution molecular dynamics simulations,
2. implementation of ITO using a denoising diffusion probabilistic model (DDPM) [4] with strong empirical results across resolutions: **SE(3)-equivariant ITO model (SE3-ITO)** gives stable long time-scale simulations and self-consistent dynamics across multiple time-scales for molecular benchmarks and **Coarse-grained SE3-ITO model (CG-SE3-ITO)** trained on large-scale protein folding data sets shows quantitative agreement with major dynamic and stationary observables of interest.

## 2 Background and Preliminaries

**Notation** Throughout this work, diffusion time, related to Diffusion Models (see Sec. 2), and physical time are represented using superscripts and subscripts, respectively.

**Molecular dynamics and observables** Molecular dynamics (MD) is a wide-spread simulation strategy in computational chemistry and physics. In this approach, the time-evolution of $N$ particles configuration in Euclidean space $\mathbf{x} \in \Omega \subset \mathbb{R}^{3M}$, is modeled via a stochastic differential equation (SDE) with a drift term based on a potential energy model $U(\mathbf{x}) : \Omega \to \mathbb{R}$. An important aim of MD is to compute:

1. **Stationary observables:** $O_f = \mathbb{E}_\mu[f(\mathbf{x})]$
2. **Dynamic observables:** $O_{f(t)h(t+\Delta t)} = \mathbb{E}_{\mathbf{x}_t \sim \mu}[\mathbb{E}_{\mathbf{x}_{t+\Delta t} \sim p_\tau(\mathbf{x}_{t+\Delta t}|\mathbf{x}_t)}[f(\mathbf{x}_t)h(\mathbf{x}_{t+\Delta t})]]$

where $\mu$ is the normalized Gibbs measure, and $p_\tau(\mathbf{x}_{t+\Delta t} \mid \mathbf{x}_t)$ is a conditional probability density function encoding the time-discrete evolution of the molecular system $\mathbf{x}$, with time-step $\Delta t = N\tau$ as prescribed by a dynamic model, e.g. *Langevin dynamics* [3], integrated with time-step, $\tau$. $N$ is typically a large integer. The functions $f, h : \Omega \to \mathbb{R}$ are observable functions or '*forward models*' describing the microscopic observation process, e.g. computing a distance or an angle. The observables, $O_{f(t)}$, and $O_{f(t)h(t+\Delta t)}$, include binding affinities and binding rates of a drug to a protein, respectively. Conventionally, these observables are estimated from simulation trajectories using naive Monte Carlo estimators.

For illustrative purposes, we assume the temporal behavior of a state, $\mathbf{x}$, follows the Brownian dynamics SDE (Itô form)

$$\mathrm{d}\mathbf{x}_t = -\nabla U(\mathbf{x}_t)\gamma^{-1}\,\mathrm{d}t + \sqrt{2D}\mathrm{d}W, \tag{1}$$

where $D = \gamma^{-1}\beta^{-1}$ is a diffusion constant, with friction $\gamma$ and inverse-temperature $\beta$, and $\mathrm{d}W$ is a Wiener process. Using the Euler–Maruyama time-discretization, with time-step $\tau$, simulating the SDE corresponds to simulating a Markov chain with the transition probability density

$$p(\mathbf{x}_{t+\tau} \mid \mathbf{x}_t, \tau) = \mathcal{N}(\mathbf{x}_{t+\tau}|\mathbf{x}_t - \tau\nabla U(\mathbf{x}_t)\gamma^{-1}, \tau\sqrt{2D}\mathbb{I}_{3M}) \tag{2}$$

where $\mathcal{N}$ specifies the multi-variate Normal distribution, and $\mathbb{I}_{3M}$ is the $3M$-dimensional identity matrix. If $\tau$ is sufficiently small to allow stable simulation, the *invariant measure*, of the Markov chain

(eq. 2), is the Boltzmann distribution (normalized Gibbs measure) corresponding to the potential energy model $U(\mathbf{x})$ at $\beta$. Consequently, by simulating a large number of steps we can draw samples from $\mu$ to compute stationary observables and compute dynamic observables by simulating $\Delta t = N\tau$ steps enough times with initial states distributed according to $\mu$. Explicit simulation make such computations extremely costly, and consequently, there's much interest in speeding up the calculations of these quantities.

**Transfer Operators**    Let $\rho$ specify an initial condition, a probability density function on $\Omega$. We can define a Markov operator $T_\Omega : L^1(\Omega) \to L^1(\Omega)$ using a transition density (e.g., 2):

$$[T_\Omega \circ \rho](\mathbf{x}_{t+\tau}) \triangleq \frac{1}{\mu(\mathbf{x}_{t+\tau})} \int_{x_t} \mu(\mathbf{x}_t)\rho(\mathbf{x}_t)p(\mathbf{x}_{t+\tau} \mid \mathbf{x}_t)\mathrm{d}\mathbf{x}_t \tag{3}$$

which then describes the $\mu$-weighed evolution of absolutely convergent probability density functions on $\Omega$ according to eq. 1, with time-step, $\tau$. Such an operator is called the (Ruelle) Transfer Operator [5, 6]. We can express the operator using a spectral form

$$T_\Omega(\tau) = \sum_{i=0}^{\infty} \lambda_i(\tau)|\psi_i\rangle\langle\phi_i| \tag{4}$$

where only eigenvalues $\lambda_i(\tau) = \exp(-\tau\kappa_i)$ depend on the time-step, $\tau$. $\kappa_i$ are characteristic 'relaxation' rates associated the left and right eigenfunction pair, $\phi_i$ and $\psi_i$ [7]. We can compute the operator with time-lag $N\tau$ via the Chapman-Kolmogorov equation (see Sec. A.1, for details)

$$T_\Omega(N\tau) = \sum_{i=0}^{\infty} \lambda_i(\tau)^N|\psi_i\rangle\langle\phi_i|. \tag{5}$$

**Equivariant Message Passing Neural Networks**    In this work, we are concerned with MD, where the time-evolution of a molecule is governed by a force field $\mathcal{F}(\cdot) \triangleq -\nabla U(\cdot)$ derived from a central potential $U(\cdot)$. While $U(\cdot)$ is *invariant* to group-actions of the Euclidean group in three dimensions (E(3)), its corresponding force field is E(3)-*equivariant*. We call a function, $f$ '*invariant*' under a group-action $g$ iif $f(\mathbf{x}) = f(S_g\mathbf{x})$ and '*equivariant*' iff $T_g f(\mathbf{x}) = f(S_g\mathbf{x})$, where $S_g$ and $T_g$ are linear representations of the group element $g$ [8].

The force field $\mathcal{F}(\cdot)$ is equivariant under E(3) group-actions. However, in practice, classical molecular dynamics simulations do not change parity during simulation, and consequently, our data distribution only contains a single mirror image of molecules.

We extended the PaiNN architecture [9], an E(3)-equivariant message passing neural network (MPNN), making it SE(3) equivariant by breaking its symmetry with respect to parity. We introduced this minor modification as we experienced sporadic parity changes when sampling with a model trained using the PaiNN architecture, and introducing this modification resolved the issue. Briefly, PaiNN embeds a graph $G = (V, E)$, where nodes, $V$, exchange equivariant messages through edges within a local neighborhood defined as $\mathcal{N}(i) = \{j \mid \|r_{ij}\| \leq r_{\text{cutoff}}\}$, where $r_{ij}$ is the distance between nodes denoted $i$ and $j$, and $r_{\text{cutoff}}$ is the maximal distance at which nodes are allowed to exchange messages. Messages are pooled and subsequently used to update node features, thereby enabling exchange of equivariant information. We achieve parity symmetry-breaking by constructing the equivariant messages in a manner that depends on cross-products between equivariant node features and direction vectors between interacting nodes. The cross-product is an axial vector (i.e., does not change sign under parity). We combine these vectors with polar vectors (change sign under parity). We refer to this modified PaiNN architecture as ChiroPaiNN (CPaiNN). Further details are in the Appendix D.

**Diffusion Models**    The diffusion model (DM) formalism is a powerful generative modeling framework that learns distributions by modeling a gradual denoising process [4, 10, 11]. In DMs, we pre-specify a *forward diffusion process* (noising process), which gradually transforms the data distribution $p(\mathbf{x}^0)$ to a simple prior distribution $p(\mathbf{x}^T)$, e.g., a standard Gaussian, through a time-inhomogenous Markov process, described by the following SDE (Itô form)

$$\mathrm{d}\mathbf{x}^t = f(\mathbf{x}^t, t)\,\mathrm{d}t + g(t)\,\mathrm{d}W. \tag{6}$$

where $0 < t < T$ is the *diffusion time*, $f$ and $g$ are chosen functions, and $\mathrm{d}W$ is a Wiener process. We can generate samples from the data distribution $p(\mathbf{x}^0)$ by sampling from $p(\mathbf{x}^T)$ and solving the *backward diffusion process* (denoising process)

$$\mathrm{d}\mathbf{x}^t = \left[ f(\mathbf{x}^t, t) - g^2(t) \nabla_{\mathbf{x}^t} \log p(\mathbf{x}^t \mid t) \right] \mathrm{d}t + g(t)\, \mathrm{d}W \tag{7}$$

by approximating the *score field* $\nabla_{\mathbf{x}^t} \log p(\mathbf{x}^t \mid t)$ — or equivalently a time-dependent Normal transition kernel [4] — with a deep neural network surrogate $\nabla_{\mathbf{x}^t} \log \hat{p}(\mathbf{x}^t \mid t, \boldsymbol{\theta})$. We can use the learned score field to define a neural ordinary differential equation (ODE) [12, 13], or probability flow ODE [14] — eq. 7 less the term $g(t)\, \mathrm{d}W$ and scaling $g^2(t)$ by $1/2$ — which we can leverage for efficient sampling and sample likelihood evaluation.

Here, we are concerned with building equivariant probability density functions under SE(3) group actions. Consequently, we parameterize the DM using a learned Normal transition kernel of a time-inhomogenous diffusion process. By restricting the transition kernels $p(\mathbf{x}^{t+1} \mid \mathbf{x}^t)$ to be equivariant under SE(3) group-actions, the marginal of $\mathbf{x}^{t+1}$ is always invariant [15]. Combining the equivariant transition kernel with an invariant prior density [16] ensures the whole Markov process is invariant to SE(3) group actions. Consequently, combining an isotropic mean-free Gaussian as prior with ChiroPaiNN-parameterized transition kernels, we can construct an SE(3) equivariant diffusion model.

Figure 2: **ITO $\hat{\epsilon}$ networks** (A) SE3-ITO used for molecular application (B) MB-ITO, used for experiments with the Müller-Brown potential. $\Lambda_{\text{pos}}$ and $\Lambda_{\text{nom}}$ are positional and nominal embedding respectively, Concat is a concatenation, and MLP is a multi-layer perceptron. Arrows are annotated with input and output shapes.

## 3   Implicit Transfer Operator

Molecular simulations are Markovian with transition density (e.g. eq. 2) and Normal, however, the latter only for very small *physical* time-steps $\tau$. Here, we aim to approximate the long-time step transition probability $p_{N\tau}(\mathbf{x}_{N\tau} \mid \mathbf{x}_0)$ to allow for one-step sampling of long-time-scale dynamics.

As data we consider simulation trajectories. The trajectories are generated by explicit simulation which corresponds to sampling ancestrally from the small time-step transition density: $\mathbf{X} = \{\mathbf{x}_\tau, \ldots, \mathbf{x}_{N\tau}\} \sim p(\mathbf{x}_{n\tau} \mid \mathbf{x}_{(n-1)\tau})$, with $n = \{1, \ldots, N\}$. In general, the state variable $\mathbf{x}$, contains both position and velocity information of the particles, along with other details such as box dimensions, depending on the simulation scheme and target ensemble. Throughout this study, we only consider the position information.

We build a surrogate of the conditional transition probability distribution — $p_{N\tau}(\mathbf{x}_{N\tau} \mid \mathbf{x}_0)$ — from MD data. In practice, we learn a generative model $\mathbf{x}_{t+N\tau} \sim p_{\boldsymbol{\theta}}(\mathbf{x}_{t+N\tau} \mid \mathbf{x}_t, N)$ with a conditional denoising diffusion probabilistic model (cDDPM) of the form

$$p(\mathbf{x}_{t+N\tau}^0 \mid \mathbf{x}_t, N) \triangleq \int p(\mathbf{x}_{t+N\tau}^{0:T} \mid \mathbf{x}_t, N)\, \mathrm{d}\mathbf{x}^{1:T} \tag{8}$$

where $\mathbf{x}^{1:T}$ are *latent variables* of the same dimension as our output, and follow a joint density describing the backward diffusion process (eq. 7) and $\mathbf{x}^T \sim \mathcal{N}(0, \mathbb{I})$. We define a conditional sample likelihood as

$$\ell(\mathbf{I}; \theta) \triangleq \prod_{i \in \mathbf{I}} p_{\boldsymbol{\theta}}(\mathbf{x}_{t_i + N_i \tau}^0 \mid \mathbf{x}_{t_i}, N_i) \tag{9}$$

where $\mathbf{I}$ is a list of generated indices $i$ specifying a time $t_i$ and a time-lag ($\tau$) integer multiple $N_i$, associating two time-points in the trajectory, $\mathbf{X}$. Following Ho et al., we train the cDDPM by

optimizing a simplified form of the variational bound of the log-likelihood [4],

$$\mathcal{L}(\theta) = \mathbb{E}_{i\sim\mathbf{I}, \epsilon\sim\mathcal{N}(0,\mathbb{I}), t_{\text{diff}}\sim\mathcal{U}(0,T)} \left[ \|\epsilon - \hat{\epsilon}_{\boldsymbol{\theta}}(\widetilde{\mathbf{x}}_{t_i+N_i\tau}^{t_{\text{diff}}}, \mathbf{x}_{t_i}, N_i, t_{\text{diff}})\|_2 \right], \tag{10}$$

where $\widetilde{\mathbf{x}}_t^{t_{\text{diff}}} \triangleq \sqrt{\bar{\alpha}^{t_{\text{diff}}}}\mathbf{x}_t + \sqrt{1-\bar{\alpha}^{t_{\text{diff}}}}\epsilon$, with $\bar{\alpha}^{t_{\text{diff}}} = \prod_i^{t_{\text{diff}}}(1-\beta_i)$ and $\beta_i$ is the variance of the forward diffusion process at diffusion time, $i$. $\hat{\epsilon}_{\boldsymbol{\theta}}(\cdot)$ is one of the two ITO neural network model architectures shown in Fig. 2, and is directly related to the score [4].

If the data used to train the conditional transition density is generated by MD simulation with time-invariant potential energy (drift), we can express the generating transition probability as a decomposition of time-variant and -invariant parts (Proof, see Sec. A.2)

$$p(x_{N\tau} \mid x_0) = \sum_{i=1}^{\infty} \underbrace{\lambda_i^N(\tau)}_{\text{time-variant}} \underbrace{\alpha_i(x_{N\tau})\beta_i(x_0)}_{\text{time-invariant}} \tag{11}$$

where $\alpha_i$ and $\beta_i$ are *time-invariant* projection coefficients of the state variables on-to the left and right eigenfunctions $\phi_i$ and $\psi_i$, of the *Transfer operator* $T_\Omega(\tau)$ [5] and $|\lambda_i(\tau)| \leq 1$ is its $i$'th eigenvalue. Consequently, we call our surrogate modeling approach *implicit transfer operator* learning.

As outlined in Algorithm 1, we generate the indices $i$, e.g. the tuples $(x_{t_i}, x_{t_i+N_i\tau}, N_i)$, in a manner such that the model is exposed to multiple time-lags, sampled uniformly across orders of magnitude, used for gradient-based optimization with Adam [17]. As a result, as illustrated in eq. 11, the model will be exposed to multiple different linear combinations of the eigenfunctions of $T_\Omega(\tau)$ in each batch during training. We conjecture that this data augmentation procedure will enable better learning of implicit representations of these eigenfunctions and, consequently, better generalization across time scales and yield more stable sampling.

### 3.1 ITO Architectures

We present two architectures for learning cDDPMs encoding ITO models, one for molecular applications SE3-ITO and one for the Müller-Brown benchmark system (Fig. 2). The SE3-ITO architecture uses our new SE(3) equivariant MPNN (ChiroPaiNN, described in sec. 2) to encode $\mathbf{x}_t$, $N$, and atom-types, $z$, to invariant features, $s$, and equivariant features, $v$. We concatenate $s$ with an encoding of the diffusion-time $t_{\text{diff}}$ and process them through a MLP (multi-layer perceptron). The output from the MLP are passed along with $v$ and $\widetilde{\mathbf{x}}_t^{t_{\text{diff}}}$ as input to a second ChiroPaiNN module which predicts $\hat{\epsilon}$. More details on the architecture and hyperparameters are available in Appendices D and E.

---

**Algorithm 1** Training. DisExp is defined in Appendix E

**Input:** $n$ MD-trajectories; $\mathcal{X} = \{\mathbf{x}_0^j, \ldots, \mathbf{x}_{t_j}^j\}_{j=0}^n$, ITO score-model; $\hat{\epsilon}_\theta$, max lag; $N_{\text{max}}$
$\mathcal{X}' = \text{Concatenate}(\{\mathbf{x}_0^j, \ldots, \mathbf{x}_{t_j-N_{\text{max}}}^j\}_{j=0}^n)$
**while** not converged **do**
  $\mathbf{x}_t \sim \text{Choice}(\mathcal{X}')$
  $N \sim \text{DisExp}(N_{\text{max}})$
  $t_{\text{diff}} \sim \text{Uniform}(0, T)$
  Take gradient step on:
  $\nabla_\theta \left[ \|\epsilon - \hat{\epsilon}_\theta(\widetilde{\mathbf{x}}_{t+N\tau}^{t_{\text{diff}}}, \mathbf{x}_t, N, t_{\text{diff}})\|_2 \right]$
**end while**
**return** $\hat{\epsilon}_\theta$

---

**Algorithm 2** Ancestral sampling. Sampling from $p_\theta$ is defined in Appendix E, Algorithm 4

**Input:** initial condition $\mathbf{x}_0$, lag $N$, ancestral steps $n$.
**Allocate** $\mathcal{T} \in \mathbb{R}^{(n+1)\times\dim(\mathbf{x}_0)}$
$\mathcal{T}[0] = \mathbf{x}_0$
**for** $i = 1 \ldots n$ **do**
  $\mathbf{x}_i \sim \hat{p}_\theta(\mathcal{T}[i-1], N)$
  $\mathcal{T}[i] = \mathbf{x}_i$
**end for**
**return** $\mathcal{T}$

---

## 4 Long time-step stochastic dynamics with Implicit Transfer Operators

### 4.1 Datasets and test-systems

To evaluate how robustly ITO models can model long time-scale dynamics, we conducted three classes of experiments, ranging from fully observed, high time-resolution, to sparsely observed and

Table 1: **VAMP2 score-gaps.** Difference in VAMP2-scores of ancestral sampling from ITO models with fixed lag and stochastic lags, compared to baseline Langevin simulations. Perfect match is 0, negative and negative values correspond to under and over estimation of meta-stability, respectively. Standard deviations on last decimal place are given in parentheses.

| system \ lag | 10 | 100 |
|---|---|---|
| Müller-Brown (fixed) | -0.0351 (5) | -0.1189 (2) |
| Müller-Brown (stochastic) | **-0.0312** (4) | **-0.0970** (5) |

low time resolution. Details on training and computational resources are available in Appendices E and F, respectively.

**Müller-Brown** is a 2D potential commonly used for benchmarking molecular dynamics sampling methods. We generate a training data-set by integrating eq. 1 with the Müller-Brown potential energy as $U(\mathbf{x})$ (For details, see Appendix B.1). This dataset corresponds to a fully observed case.

**Alanine dipeptide** We use publicly available data from MDshare [18]. Simulation is performed with $2\,\mathrm{fs}$ integration time-steps and data is saved at $1\,\mathrm{ps}$ intervals. The simulations are performed in explicit solvation, but we only model the 22 atoms of the solute, without considering velocities. Consequently, this dataset is only partially observed.

**Fast-folding proteins** We use molecular dynamics data previously reported by Lindorff-Larsen et al. on the fast-folding proteins Chignolin, Trp-Cage, BBA, and Villin [19]. The data is proprietary but available upon request for research purposes. The simulations were performed in explicit solvent with a $2.5\,\mathrm{fs}$ time-step and the positions was saved at $200\,\mathrm{ps}$ intervals. We coarse-grain the simulation by representing each amino-acid by the Euclidean coordinate of their $C\alpha$ atom as done previously [20], leading to 10, 20, 28, and 35 particles in each system respectively. Consequently, these data correspond to a mostly unobserved case.

### 4.2 Stochastic lag improves meta-stability prediction

In sec. 2, we conjecture that exposing an ITO model to multiple lag times during training leads to better and more robust models. To test this, we trained a set of models on the Müller-Brown dataset with fixed constant lags $N = \{10, 100, 1000\}$ (fixed lag) and a single model with $N \sim \mathrm{DisExp}(1000)$ (stochastic lag) using the MB-ITO model (Fig. 2).

We find that the model trained with a stochastic lag systematically outperforms models trained with fixed lag (Table 1). We gauge the agreement by comparing Variational Approach to Markov Processes (VAMP)-2 scores [21] (for details, see Appendix G), between model samples and training data and find that both models tend to underestimate meta-stability compared to training data slightly. However, the model trained with stochastic lag is marginally closer to the reference values. We note that the difference in the ability of fixed and stochastic lag ITO models to capture long-time-scale dynamics is also reflected in the learned transition densities (Fig. 5). Together, these results suggest that lag-time augmentation during training leads to better implicit learning of the Transfer operator's eigenfunctions than training with a fixed lag.

To test whether this phenomena extends in cases where we do not have full observability and to molecular systems we followed the VAMP2-gaps of alanine di-peptide as a function of epoch for models trained with a fixed lag and a stochastic lag (Fig. 6). We find that the VAMP-2 gaps for stochastic lag and fixed lag in this case are statistically indistinguishable across all epochs. These results suggest that we can without compromising on accuracy build multiple time-scale surrogates by training with stochastic lag-times.

### 4.3 Efficient and accurate self-consistent long time-scale dynamics

We evaluate the ITO models trained with stochastic lags to capture long time-scale dynamics in a self-consistent manner, in the Chapman-Kolmogorov sense, i.e., $p(\mathbf{x}_{\Delta t} \mid \mathbf{x}_0) \triangleq p(\mathbf{x}_{N\tau} \mid \mathbf{x}_0) = \prod_{i=1}^{N} p(\mathbf{x}_{i\tau} \mid \mathbf{x}_{(i-1)\tau})$, or if samples generated by direct sampling with time-step $\Delta t = N\tau$ are

distributed similarly to samples generated by performing ancestral sampling $N$ times, each with time-step $\tau$. In direct sampling, we draw samples for a desired time-step $\Delta t = N\tau$ from $\hat{p}_\theta(\mathbf{x}_0, N)$ and in ancestral sampling we draw $n$ samples with time-step $\Delta t = N\tau$ from $\hat{p}_\theta(\mathbf{x}_0, N)$ in an ancestral manner, e.g. $x_{i+1} \sim \hat{p}_\theta(\mathbf{x}_i, N)$, where $i = \{0, \ldots n-1\}$.

For the fully-observed Müller-Brown case, we find that the ITO model is self-consistent by the strong overlap in transition densities sampled in a direct and ancestral manner (Algorithm 2). These results generalize to molecular systems and partially observed systems. Sampling an SE3-ITO model (Fig. 2) trained with alanine dipeptide data, we find strong agreement between the ancestrally and directly sampled transition densities (Fig. 3) and we again have a strong consistency with corresponding transition densities computed from molecular dynamics simulations. Note here, that the time-step of the ITO-sampled transition densities varies from $10^4$ to $10^6$ times the MD integration time-step. The transition densities for alanine di-peptide (Fig. 3) are calculated using 15'000 trajectories. For direct sampling, this means that we draw 15'000 samples in total. In the case of ancestral sampling, we sample 4, 64, or 512 steps for 15'000 trajectories, to match $\Delta t$.

Next, we consider four fast-folding proteins [19] where we coarse-grain the proteins by representing them only with their $C\alpha$ atoms. In this sparsely observed case (CG-SE3-ITO), we find strong model self-consistency, as shown by the comparison between conditional densities from the folded and unfolded states (Fig. 4) projected onto a linear subspace determined using *time-lagged independent component analysis* (time-lagged independent components, tIC) [22] (see Appendix B.3). Further, the long time-scale transition density gradually converges to the data distribution as expected.

Finally, by ancestral sampling (Algorithm 2), we perform a simulation of Chignolin with the same length as the training trajectory ($106 \, \mu s$), using a CG-SE3-ITO model, and compare with MD. The CG-SE3-ITO simulation is 2120 steps with $\Delta t = 5 \, \mathrm{ns}$. Running in parallel, on a single Titan X GPU we can simulate the CG-SE3-ITO model at a rate of $363 \, \mathrm{ns}/(\mathrm{s}_w M^2)$ where $\mathrm{s}_w$ denotes seconds wall-time (Appendix C.1). Remarkably, these trajectories are virtually indistinguishable in the slowly

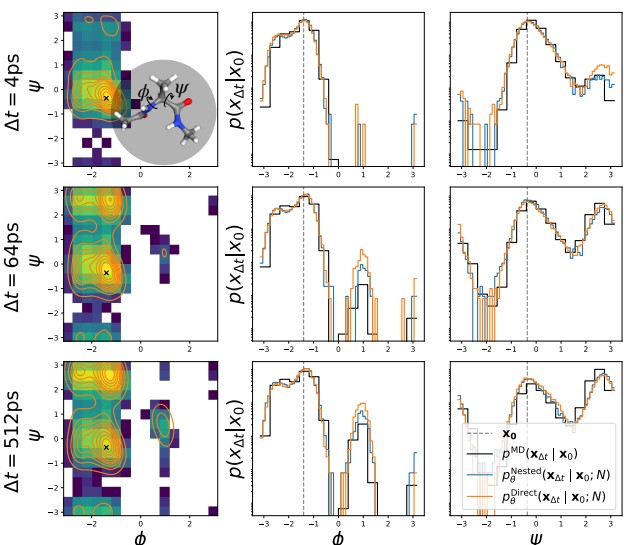

Figure 3: **Transition probability densities of alanine dipeptide dynamics with SE3-ITO model**; Rows of increasing time-lag (from top to bottom). Contours are samples from SE3-ITO model, and 2D histograms show estimates from MD data. The first column shows conditional transition densities projected onto the torsion angles $\phi$ and $\psi$ (inset). The black cross indicates the initial condition. The second and third columns show marginal distributions of $\phi$ and $\psi$, respectively, with direct sampling in orange, ancestral sampling in blue, and MD data in black.

relaxing TICA coordinates, illustrating stability of ITO. These conclusions extend to the proteins Villin, BBA, and Trp-Cage (See Appendix, Figs. 8, 9 and 10)

Together these results suggest that ITO models accurately and self-consistently capture the slow dynamics of molecular systems and are robust to situations where the system is only partially observed. In general, we expect robustness to sparsely observed representations as long as the input representations are sufficient to span the eigenfunctions of $T_\Omega$ [23, 24]. Approximation errors will translate into systematic under-estimation of relaxation time-scales [7], consistent with our slight under-estimation of VAMP-2 scores (Table 1). In future work, combining the learning of SE3-ITO models with a systematic scheme for coarse-graining [25, 26], could be an avenue for scaling to large-scale molecular systems at a low computational cost.

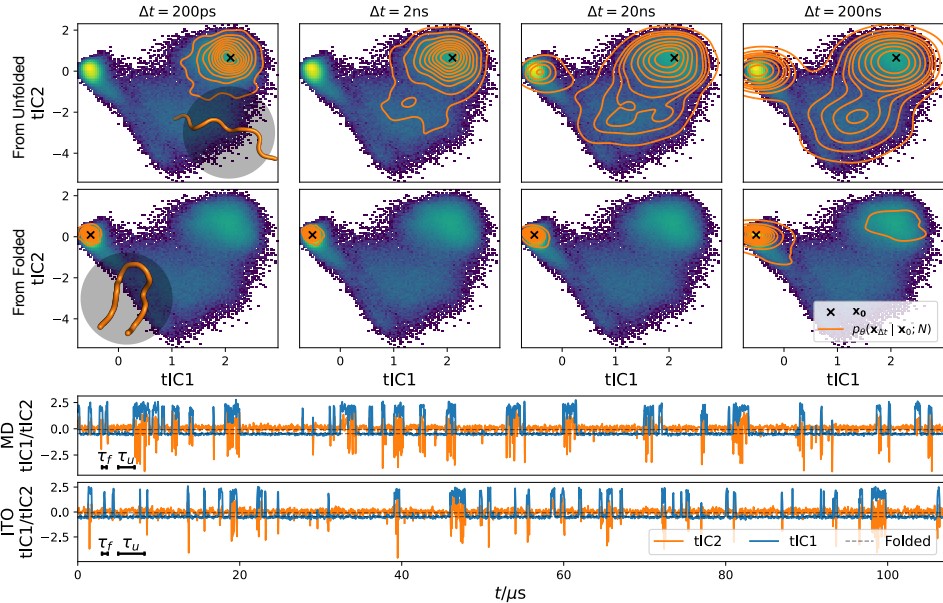

Figure 4: **Reversible protein folding-unfolding of Chignolin with CG-SE3-ITO** Conditional probability densities (orange contours) starting from unfolded (upper panels) and folded (lower panels) protein states, at increasing time-lag (left to right), shown on top of data distribution. Below: time-traces of 106 microsecond MD simulations and ITO simulations on tICs 1 and 2. The two dashed lines correspond to the folded state value in tIC 1 (lower line) and tIC 2 (higher line). Contour lines are based on 10'000 trajectories, generated with ancestral sampling with the length, $\Delta t$ and time-step 200 ps. For $\Delta t = 200\,\text{ns}$ this corresponds to 1000 ancestral samples.

## 5   Prediction of dynamic and stationary observables of using CG-SE3-ITO

As outlined in section 2, an important aim of MD simulations is to compute stationary and dynamic observables, which involves intractable integrals typically approximated via Monte Carlo estimators. Using the trained ITO models we can efficiently sample i.i.d. from the transition density needed for computing dynamic observables, and by choosing a time-step which is sufficiently large we can also sample i.i.d from the Boltzmann distribution $\mu$, the latter akin to *Boltzmann generators* [27] (See Appendix A.1). We note that, the ITO models are surrogates and as such without reweighing we cannot expect unbiased samples from the Boltzmann and dynamic transition densities. Nevertheless, we gauge how accurately ITO models we can compute these observables of interest in the context of protein folding without reweighing:

- **Free Energy of Folding**, $\Delta G = -\log\left[\frac{p_{\text{f}}}{1-p_{\text{f}}}\right]$

- **Mean first passage time, folding**, $\langle\tau_f\rangle = \int_{x_0\in\neg f}\int_0^\infty \delta(x_t\in f)p(x_t\mid x_0,t)t\,\mathrm{d}t\,\mathrm{d}x_0$

- **Mean first passage time, unfolding**, $\langle\tau_u\rangle = \int_{x_0\in f}\int_0^\infty \delta(x_t\in\neg f)p(x_t\mid x_0,t)t\,\mathrm{d}t\,\mathrm{d}x_0$

where $\{f,\neg f\}\subset\Omega$ are disjoint subsets corresponding to the folded and unfolded states of a protein, $p_f = \int_{x\in f}\mu(x)\,\mathrm{d}x$, is the folded state probability and $\delta(\cdot)$ is the Dirac delta.

We compute these observables using the reference molecular simulation data [19] and sample statistics from the CG-SE3-ITO models of each of the four fast-folding proteins (details in Appendix B.6). Strikingly, the observables computed using CG-SE3-ITO models agree well with those computed from long all-atom MD simulations (Table 2).

Finally, we analyzed the robustness, convergence, and consistency of these observables (Fig. 7). For Chignolin, we trained five models independently and analyzed model checkpoints when the training loss had stabilized. For each checkpoint and each model, we computed the observables. The values predicted are statistically indistinguishable, suggesting consistency, robustness, and convergence. The

Table 2: **Molecular observables** Standard deviations on last decimal place are given in parentheses. Stationary and dynamic observables are denoted **s** and **d**, respectively.

| | $\Delta G_{\text{fold}}/kT$ (**s**) | $\langle\tau_f\rangle/\mu s$ (**d**) | $\langle\tau_u\rangle/\mu s$ (**d**) |
|---|---|---|---|
| Chignolin (MD/ITO) | -1.28(1)/-0.64(33) | 0.565(4)/1.02(24) | 2.01(2)/2.12(34) |
| Trp-Cage (MD/ITO) | 1.47(6)/2.84(6) | 13.6(4)/37(2) | 3.4(2)/2.85(9) |
| BBA (MD/ITO) | 0.97(3)/1.52(3) | 11.7(2)/8.6(2) | 5.1(1)/1.75(4) |
| Villin (MD/ITO) | 1.21(2)/2.22(3) | 2.41(3)/3.27(7) | 0.68(1)/0.354(5) |

average predictions closely match the reference values. Nevertheless, we note that the fluctuations in these values are noticeable.

We implemented all experiments using PyTorch [28], PyTorch Lightning [29], JAX [30], and used DPM-Solver [31] for probability flow ODE Sampling.

## 6 Related Work

**Molecular sampling**   Sampling molecular configurations is a broad field and can broadly be divided into two main areas: physically motivated sampling of the Boltzmann distribution and conformer generation. The first area includes algorithmic approaches to sample the Boltzmann distribution including Molecular Dynamics simulations [2], Markov Chain Monte Carlo, extended ensemble methods [32, 33, 34, 35], including analysis methods involving deep generative nets [36], and surrogate models which directly approximate the Boltzmann distribution and allow for recovery of unbiased statistics, including Boltzmann generators [37, 16, 38, 39]. Conformer generation concerns generating physically plausible conformers without explicitly trying to follow the Boltzmann distribution. The latter approaches can be split into ML [15, 40, 41, 42, 43, 44, 45, 46, 47, 48, 49] and chemoinformatic [50, 51] approaches. Finally, speeding up molecular simulations by reducing the effective number of particles to simulate through coarse-graining with special purpose forcefield models [52] including machine learned variants [53, 54, 20, 55] and learned coarse-graining maps [25, 26] is an orthogonal approach to sample conformation space. Further, several methods to recover all-atom models from coarse-grained representations through ML [56, 57] and rule-based approaches [58] are available.

**Transfer Operator surrogates**   Building transfer operator surrogates is commonly used in molecular modeling including (Deep Generative) Markov state models (MSM) [59, 7, 60, 61], also including experimental data [62, 63] dynamic graphical models, [64] VAMPnets [65, 21], observable operator models[66], however, primarily for analysis of molecular dynamics data. Markov state models are time-space discrete approximations of the transfer operator and Deep Generative MSM [67] and VAMPnets [65] are deep learning infused versions, where state discretization is learned by deep nets. Dynamic graphical models reparameterize MSMs as kinetic Markov random fields allowing for scaling to larger systems [64]. Klein et al. recently introduced *timewarp* which is a flow-based generative model to simulate molecular systems with a large (up to $0.5\,\text{ns}$), fixed, time-lag, [68] providing asymptotically unbiased equilibrium samples through a Metropolis-Hastings correction [69]. While timewarp generates conformers with realistic local structure, it has limitations in capturing long time-scale dynamics, which is reflected in the predicted transition probability densities. In contrast, our approach captures long time-scale dynamics efficiently allowing for accurate prediction of dynamic observables. However, currently, neither the code nor the data from timewarp is publicly available percluding direct comparisons on the benchmark tasks established in their paper.

## 7 Limitations

**Surrogate model**   Implicit Transfer Operators are surrogate models of stochastic dynamics' conditional transition probability densities. We cannot guarantee unbiased sampling of dynamics and the stationary distribution due to aleatoric (e.g., finite data) and epistemic (e.g., model misspecification) uncertainty. We can overcome the latter by reweighing against a Markov Chain Monte Carlo acceptance criterion as proposed previously [68], to ensure unbiased dynamics path-reweighing is necessary, which in turn requires closed-form expressions for the target path probabilities [70].

**Transferability and scalability**   Currently, ITO does not generalize across chemical space and thermodynamic variables. In future work, we anticipate that generalization across chemical space limitations can be overcome by appropriate data set curation and parameter-sharing schemes [71]. Generalization across thermodynamic variables such as temperature and pressure would require using a surrogate model which is steerable under these changes, e.g., temperature steerable flows [72] or thermodynamic maps [73]. Currently, we assume a fully connected graph that scales $\mathcal{O}(M^2)$ in system size, which limits what systems are practically accessible. Devising new surrogate models which use mean-field approximation approaches from e.g., computational physics [74] or chemistry to truncate the graphs and treat long-range as an additive term [75] could yield more favorable scaling [76].

## 8   Conclusions

This paper introduces Implicit Transfer Operators (ITO), an approach to building multiple time-scale surrogate models of stochastic molecular dynamics. We implement ITO models with a conditional DDPM using a new time-augmentation scheme and show how ITO models capture fast and slow dynamics on benchmarks and molecular systems. We show ITO models are self-consistent over multiple time scales and highly robust to the marginalization of degrees of freedom in the system, which are unimportant to capture the long-time-scale dynamics. Combined with a SE(3) variant of the PaiNN architecture [9] (ChiroPaiNN), we further show strong empirical evidence of scaling to molecular applications, such as the folding of coarse-grained proteins. As such, we are confident that ITO is a stepping-stone toward general-purpose surrogates of molecular dynamics.

## Acknowledgments and Disclosure of Funding

This work was partially supported by the Wallenberg AI, Autonomous Systems and Software Program (WASP) funded by the Knut and Alice Wallenberg Foundation and The Novo Nordisk Foundation (SURE, NNF19OC0057822). OW's work was funded in part by the Novo Nordisk Foundation through the Center for Basic Machine Learning Research in Life Science (NNF20OC0062606) and by the Pioneer Centre for AI, DNRF grant number P1. Preliminary results were enabled by resources provided by the National Academic Infrastructure for Supercomputing in Sweden (NAISS) at Alvis (project: NAISS 2023/22-392) partially funded by the Swedish Research Council through grant agreement no. 2022-06725.

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
