# A Properties of the Transfer operator

## A.1 Relaxation of $T_\Omega$ spectrum

In this section, we outline the 'relaxation' or 'decay' of the spectral components of $T_\Omega$ as a function of time-step, $\tau$. We use that $\langle \phi_i | \psi_i \rangle_\mu = \int \phi_i(x)\psi_i(x)\,\mathrm{d}\mu(x) = 1$ if $i = j$ and $0$ if $i \neq j$, e.g. the eigenfunctions are orthonormal under the $\mu$-weighed inner-product. Since $T_\Omega(\tau)$ is Markov, composing $T_\Omega(\tau)$ with itself $N$ times we get,

$$
\begin{aligned}
[T_\Omega(\tau)]^N &= T_\Omega(\tau) \circ \cdots \circ T_\Omega(\tau) && (12) \\
&= \sum_{i=0}^{\infty} \lambda_i(\tau)|\psi_i\rangle\langle\phi_i|\lambda_i(\tau)|\psi_i\rangle\langle\phi_i|\ldots\lambda_i(\tau)|\psi_i\rangle\langle\phi_i| && (13) \\
&= \sum_{i=0}^{\infty} \lambda_i(\tau)^N|\psi_i\rangle\langle\phi_i|\psi_i\rangle_\mu\langle\phi_i|\ldots|\psi_i\rangle\langle\phi_i| && (14) \\
&= \sum_{i=0}^{\infty} \lambda_i(\tau)^N|\psi_i\rangle\langle\phi_i| && (15)
\end{aligned}
$$

We assume the dynamics governed by $T_\Omega$ are

1. reversible $\lambda_i \in \mathbb{R}$

2. measure-preserving $0 \leq |\lambda_i| \leq 1$

3. ergodic, $\lambda_0 = 1$ and $|\lambda_{i>0}| < 1$

where we have sorted the eigenvalues eigenfunction pairs in descending order. Consequently, for $N \to \infty$ we have $T_\Omega(N\tau) \to |\mathbb{1}\rangle\langle\mu|$, where $\mathbb{1}$ is the constant function.

## A.2 Decomposition of transition density

In this section, we detail the decomposition of the transition density, $p(\mathbf{x}_{N\tau} \mid \mathbf{x}_0)$.

Let $\rho$ specify an initial condition, an absolutely convergent probability density function on $\Omega$. We can define a Transfer operator $T_\Omega$ using a transition probability density [6]:

$$
[T_\Omega \circ \rho](\mathbf{x}_{N\tau}) \triangleq \frac{1}{\mu(\mathbf{x}_{N\tau})}\int_{\mathbf{x}_0} \mu(\mathbf{x}_0)\rho(\mathbf{x}_0)p(\mathbf{x}_{N\tau} \mid \mathbf{x}_0)\,\mathrm{d}\mathbf{x}_0, \qquad T_\Omega : L^1(\Omega) \to L^1(\Omega) \qquad (16)
$$

which then describes the $\mu$-weighed evolution of densities on $\Omega$ according to MD discretized in time by a step-size of $\tau$. $\mu$ is a normalized Gibbs measure, or the Boltzmann distribution.

Since we only consider MD with time-invariant drift, only the eigenvalues $\lambda_i(\tau)$ of $T_\Omega(\tau)$ depend on $\tau$. We can express arbitrary transition probabilities through a bilinear form

$$
p(\mathbf{x}_{N\tau} \mid \mathbf{x}_0) = \langle\delta_{\mathbf{x}_{N\tau}}|T_\Omega^N(\tau)|\delta_{\mathbf{x}_0}\rangle = \sum_{i=1}^{\infty}\lambda_i^N(\tau)\langle\delta_{\mathbf{x}_{N\tau}}|\phi_i\rangle\langle\psi_i|\delta_{\mathbf{x}_0}\rangle = \sum_{i=1}^{\infty}\lambda_i^N(\tau)\alpha_i(\mathbf{x}_{N\tau})\beta_i(\mathbf{x}_0)
$$

$$(17)$$

where $\alpha_i$ and $\beta_i$ are *time-invariant* projections coefficients of the state variables on-to the eigenfunctions $\phi_i$ and $\psi_i$, and $\delta_{\mathbf{x}}$ is the Dirac delta centered at $\mathbf{x}$. $T_\Omega^N(\tau)$ means $T_\Omega(\tau)$ acting $N$ times (See A.1).

# B Datasets

Throughout we train on all available data, as it is often sparse and difficult to split in an appropriate manner due to rare events e.g. folding and unfolding.

Table 3: Details about the Alanine dipeptide data (taken verbatim from mdshare)

| Property | Value |
|---|---|
| Code | ACEMD |
| Forcefield | AMBER ff-99SB-ILDN |
| Integrator | Langevin |
| Integrator time step | 2 fs |
| Simulation time | 250 ns |
| Frame spacing | 1 ps |
| Temperature | 300 K |
| Volume | $(2.3222nm)^3$ periodic box |
| Solvation | 651 TIP3P waters |
| Electrostatics | PME |
| PME real-space cutoff | 0.9 nm |
| PME grid spacing | 0.1 nm |
| PME updates | every two time steps |
| Constraints | all bonds between hydrogens and heavy atoms |

## B.1 Müller Brown

We generate the Müller Brown data set used for training by integrating the 2D potential energy model:

$$U(x,y) = \sum_{i=1}^{4} A_i \exp \left[ a_i(x - \bar{x}_i)^2 + b_i(x - \bar{x}_i)(y - \bar{y}_i) + c_i(y - \bar{y}_i)^2 \right] \tag{18}$$

using simulating overdamped Langevin or Brownian dynamics SDE, through a Euler-Mayurama time-discretization, and where

$$
\begin{aligned}
A &= (-200, -100, -170, 15) \\
a &= (-1, -1, -6.5, 0.7) \\
b &= (0, 0, 11, 0.6) \\
c &= (-10, -10, -6.5, 0.7) \\
\bar{x} &= (1, 0, -0.5, -1) \\
\bar{y} &= (0, 0.5, 1.5, 1).
\end{aligned}
\tag{19}
$$

We we generate 32 trajectories with random initial conditions in the ranges

$$
\begin{aligned}
x &= [-1.5, 1.2] \\
y &= [-0.2, 2.0],
\end{aligned}
\tag{20}
$$

and save every 10th step after a burn-in of 1000 steps. Each trajectory is simulated for 100000 steps.

A separate testing set was generated in an identical manner but with a different random seed. The values in Table. 1 are computed compared to this test set.

## B.2 Alanine dipeptide

We use the data from MDShare (Table 3) which consists of three independent trajectories of 250 ns each.

**Pre-processing** The atomic coordinates are standardized before model training, each atom has a unique nominal embedding as atom type.

## B.3 Fast folding proteins

The original data was obtained upon request from DE Shaw Research, and details about the simulations are available in the original publication [19]. All configurations were preprocessed by centering them at the origin. Furthermore, all configurations were scaled to ensure a standard deviation of one across the dataset.

Table 4: MSM hyperparameters. All models used 100 cluster centers, and clustered in the 5 first TICs.

|  | TICA lag | MSM lag | ITO $\Delta t$ |
|---|---|---|---|
| Chignolin | 1ns | 100ns | 200ns |
| Trp-Cage | 1ns | 100ns | 200ns |
| BBA | 1ns | 800ns | 200ns |
| Villin | 1ns | 200ns | 200ns |

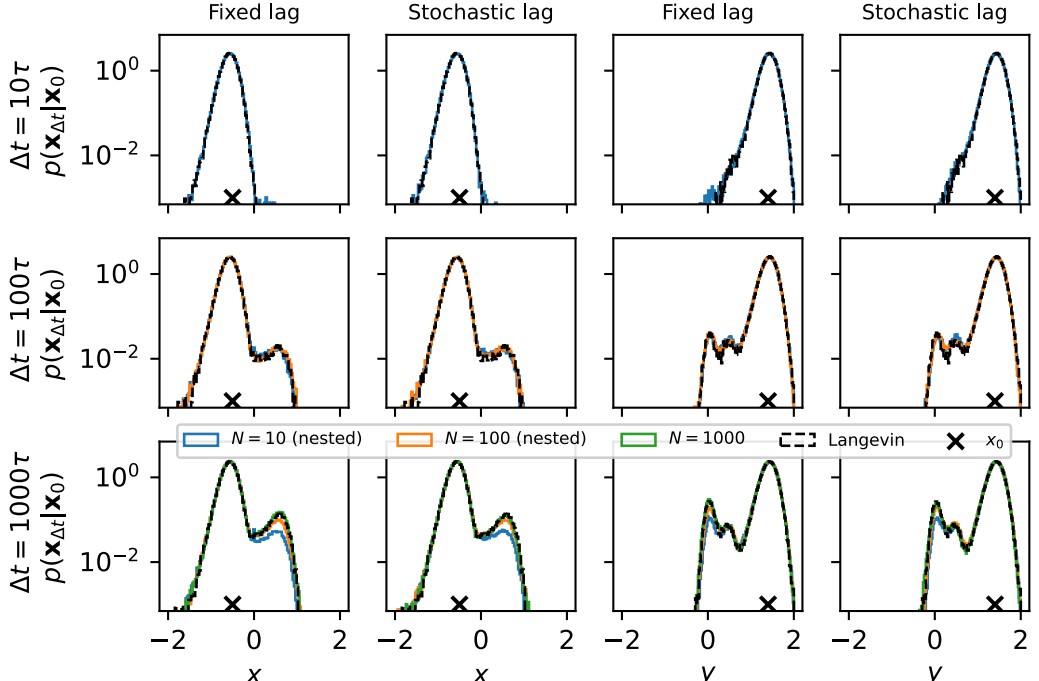

Figure 5: **Müller-Brown potential.** Conditional Probability Densities starting in $x_0$ indicated by cross, in ITO models trained with fixed or stochastic lag. Comparison of histograms of direct and ancestral sampling to direct simulation (Langevin). $N_{\text{samples}=250k}$

## B.4 Müller-Brown

## B.5 Alanine di-peptide

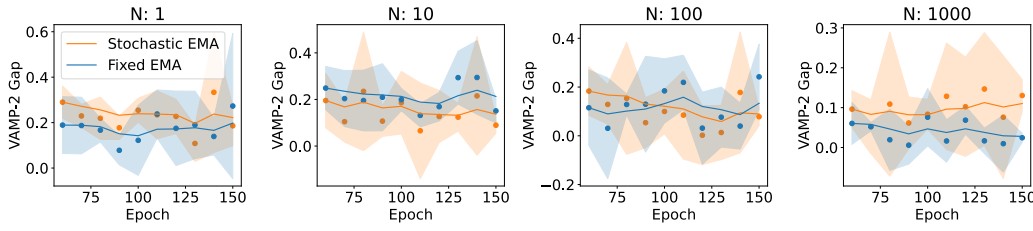

Figure 6: **VAMP-2 gap for Alanine di-peptide** The plots show exponential moving averages (EMA) VAMP-2 gaps for stochastic lag (Stochastic) and fixed lag models (Fixed) as a function of training epoch, for lags ($N$) 1, 10, 100, and 1000. Shaded areas correspond to the exponential moving standard deviation.

## B.6  Fast folding proteins

Figures 8, 9 and 10, show conditional distributions generated by CG-SE3-ITO models and comparisons of MD with ITO simulations on the fast folders Trp-Cage, BBA, and Villin, respectively.

**Reference value and observables**   We compute observables using Markov state models. First, we estimate a reference model for each system (see hyper-parameters in Table 4). Briefly, non-redundant and non-trivial pair-wise $C\alpha$ distances were used as input for TICA dimension reduction, the reduced space was clustered using k-means. MSMs were sampled from a Bayesian posterior as previously described [77], using cluster assignments as state assignments. We identified folded and unfolded states using PCCA (Perron Cluster-Cluster analysis) [78], which in turn enabled the calculation of mean first passage times (MFPT) of folding $\langle \tau_f \rangle$ and unfolding $\langle \tau_u \rangle$ and the free energy of folding $\Delta G_{\mathrm{fold}}$.

Observables computed from ITO simulations were computed by processing the simulation data by projecting them onto the TICA space and the cluster centers determined on the MD data. MSMs were sampled as for MD data and observables were computed in the same way.

The reported uncertainties are standard deviations from Bayesian posterior sampling.

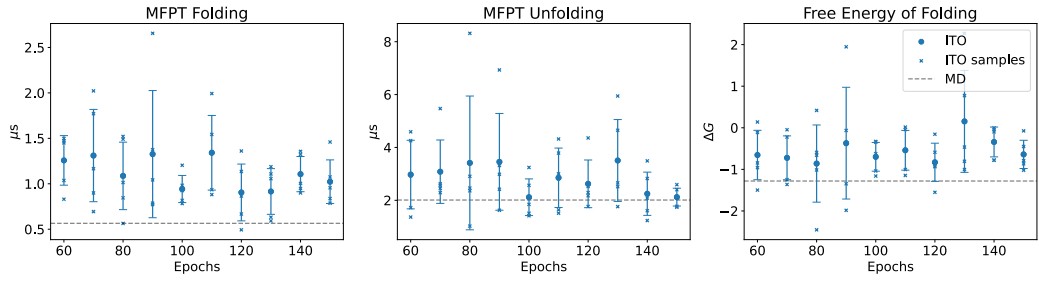

Figure 7: **Robustness, convergence, and consistency of observables in Chignolin** Average and standard deviation of observables as a function of epoch following stabilization of training loss. The averages and standard deviations are computed using samples from five independently trained models (crosses). Reference values computed from MD data are shown with dashed lines.

# C   Additional results

## C.1   Sample timings

Running on a single device of a NVIDIA TITAN V node, using all memory, we can concurrently generate

- 253 simulation-steps/s for Chignolin
- 61 simulation-steps/s for Trp-Cage
- 35 simulation-steps/s for BBA
- 21 simulation-steps/s for Villin
- 48 simulation-steps/s for Alanine-Dipeptide

Note that all samples presented in this paper have been calculated equivalently using 50 ODE-steps. Depending on simulated lag, arbitrarily long trajectories can be sampled efficiently. Our models were trained on lags of up to 200 ns, but our findings suggest no constraints on extending the framework to much longer time scale.

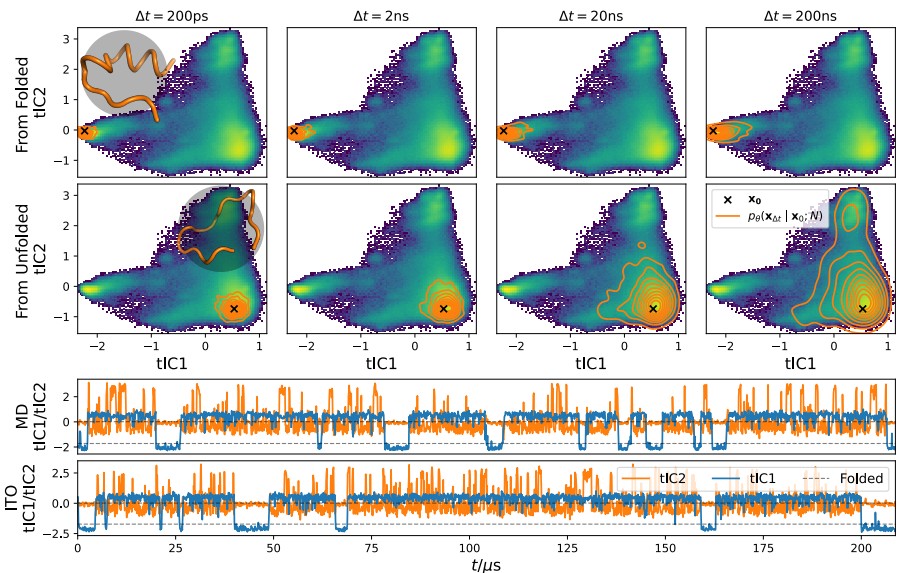

Figure 8: **Reversible protein folding-unfolding of Trp-Cage with CG-SE3-ITO** Conditional probability densities (orange contours) starting from folded (upper panels) and unfolded (lower panels) protein states, at increasing time-lag (left to right), shown on top of data distribution. Below: time-traces of 208 microsecond MD simulations and ITO simulations on tICs 1 and 2. Contour lines are based on 10'000 trajectories, generated with ancestral sampling with the length, $\Delta t$ and time-step 200 ps. For $\Delta t = 200$ ns this corresponds to 1000 ancestral samples.

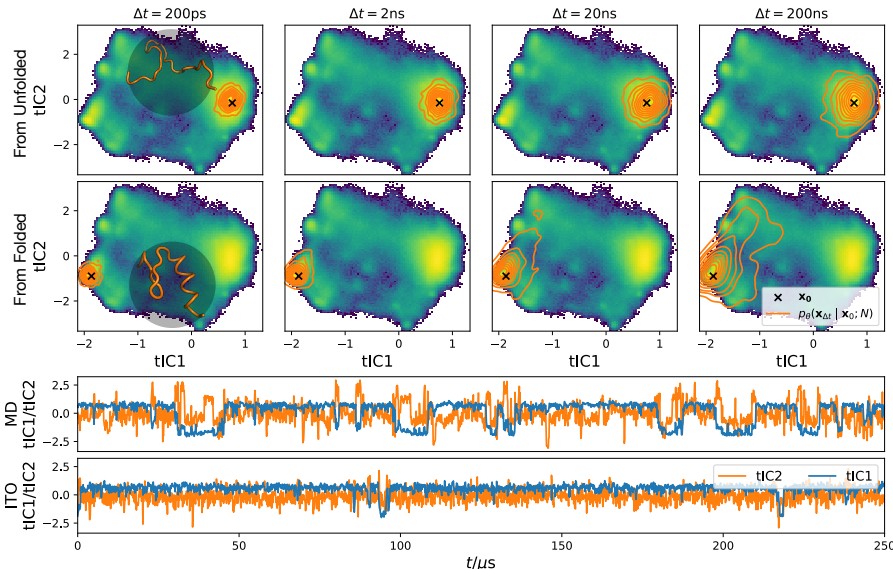

Figure 9: **Reversible protein folding-unfolding of BBA with CG-SE3-ITO** Conditional probability densities (orange contours) starting from unfolded (upper panels) and folded (lower panels) protein states, at increasing time-lag (left to right), shown on top of data distribution. Below: time-traces of 250 microsecond MD simulations and ITO simulations on tICs 1 and 2. Contour lines are based on 10'000 trajectories, generated with ancestral sampling with the length, $\Delta t$ and time-step 200 ps. For $\Delta t = 200$ ns this corresponds to 1000 ancestral samples. Contour lines are based on 10'000 trajectories, generated with ancestral sampling with the length, $\Delta t$ and time-step 200 ps. For $\Delta t = 200$ ns this corresponds to 1000 ancestral samples.

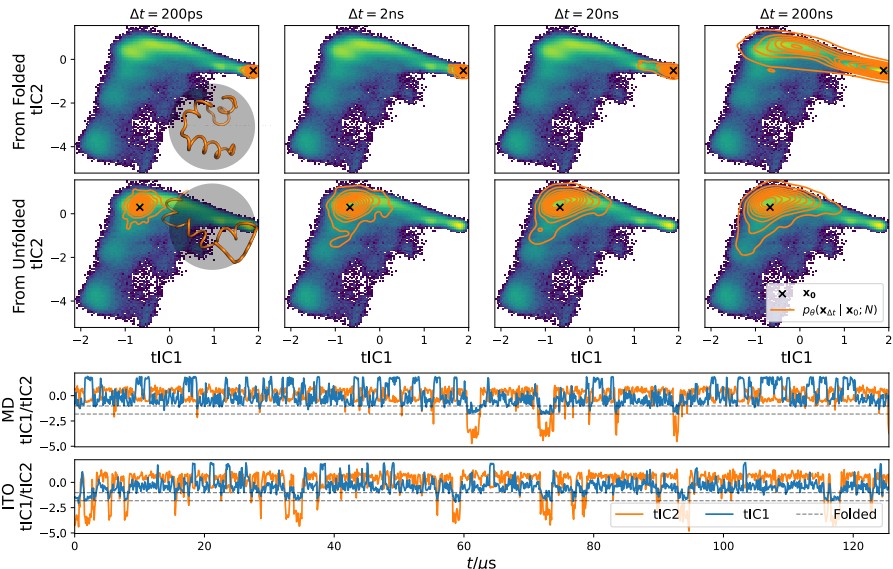

Figure 10: **Reversible protein folding-unfolding of Villin with CG-SE3-ITO** Conditional probability densities (orange contours) starting from folded (upper panels) and unfolded (lower panels) protein states, at increasing time-lag (left to right), shown on top of data distribution. Below: time-traces of 125 microsecond MD simulations and ITO simulations on tICs 1 and 2.

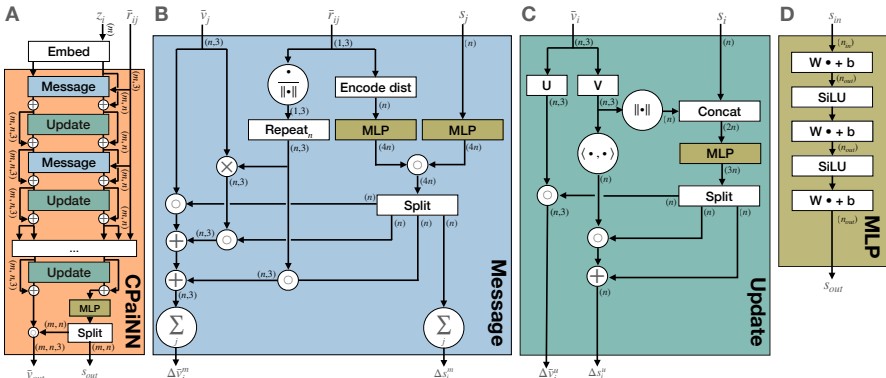

Figure 11: **ChiroPaiNN architecture** utilized in SE3-ITO and CG-SE3-ITO models (Fig. 2) for the embedding of conditional configuration and score prediction. Arrows are annotated with input and output shapes. $\times$ indicates cross product operations between all vectors along the first dimension, and $\circ$ indicates element-wise multiplication along the first dimension.

## D   Architectural details

**Positional embedding,** $\Lambda_{\mathrm{pos}}$, maps diffusion time $t_{\mathrm{diff}}$, physical time $\Delta t$, and interatomic distances $r_{ij}$ to $n$-dimensional features-vectors with the $n$'th dimension defined as:

$$\Lambda_{\mathrm{pos}}^n(x) = \begin{cases} \cos\left(\left(1 + \frac{n}{2}\right) x \frac{\pi}{l_0}\right) & \text{for even } n \\ \sin\left(\left(1 + \frac{n-1}{2}\right) x \frac{\pi}{l_0}\right) & \text{for odd } n, \end{cases} \tag{21}$$

where $l_0$ is a hyperparameter.

**Nominal embedding** $\Lambda_{\mathrm{nom}}$ , maps atomic elements or residue types to continuous $n$-dimensional feature vectors, $f : C \to R^n$, where $C$ is the set of all categorical values and $n$ is the dimension of the embedded vector.

# E   Training details

## E.1   Sampling of configurations

The last $N_{\mathrm{max}}$ frames were truncated from each trajectory such that $\mathbf{x}_t$ could be sampled uniformly while keeping $\mathbf{x}_{t+N_{\mathrm{max}}}$ in bounds. $N$ is sampled discretely from $\mathrm{DisExp}(N_{\mathrm{max}})$ following;

---

**Algorithm 3** Sampling from DisExp

$N_{\mathrm{log}} \sim \mathrm{Uniform}(0, \log(N_{\mathrm{max}}))$
**Return:** $\mathrm{floor}(\exp(N_{\mathrm{log}}))$

---

---

**Algorithm 4** Sampling from $\hat{p}_{\boldsymbol{\theta}}(\mathbf{x}_0, N)$

**Input:** initial condition $\mathbf{x}_0$, lag; $N$, diffusion steps; $T_{\mathrm{diff}}$, ITO score-model; $\hat{\epsilon}_\theta$
$\mathbf{x}_N^{T_{\mathrm{diff}}} \sim \mathcal{N}(\mathbf{0}, \mathbf{1})$
**for** $t_{\mathrm{diff}} = T_{\mathrm{diff}} \dots 1$ **do**
$\quad \epsilon \sim \mathcal{N}(\mathbf{0}, \mathbf{1})$
$\quad \mathbf{x}_N^{t_{\mathrm{diff}}-1} = \frac{1}{\sqrt{\alpha^{t_{\mathrm{diff}}}}} \left( \mathbf{x}_N^{t_{\mathrm{diff}}} - \frac{1-\alpha^{t_{\mathrm{diff}}}}{\sqrt{1-\bar{\alpha}^{t_{\mathrm{diff}}}}} \hat{\epsilon}_\theta(\mathbf{x}_N^{t_{\mathrm{diff}}}, \mathbf{x}_0, N, t_{\mathrm{diff}}) \right) + \sigma_t \epsilon$
**end for**
**return** $\mathbf{x}_N^0$

---

## E.2   Data splits

All available data was used for training with no test/validation set. Reference MFPT values are already coarse estimates and cannot be accurately calculated from a subset of the data due to slow time scales compared to the length of available trajectories.

## E.3   Hyper Parameters

**Müller-Brown**   For the Müller-Brown results we trained with the MLP in MB-ITO architecture with 32 dimensional positional embeddings for $t_{\mathrm{phys}}$ and $N$ and the MLP had 32 hidden nodes and 5 layers. We used a cosine learning rate scheduler and a sigmoidal $\beta$-scheduler with parameters as listed for alanine dipeptide and the fast folders. The model with stochastic lag was trained with $N_{\mathrm{max}} = 1000$ and for fixed lag models $N$ was fixed during data generation and the positional embeddings of $N$ were removed from the model.

**Alanine dipeptide and Fast folders**   Hyperparameters employed for experiments on the fast folding proteins and Alanine Dipeptide are outlined below:

```
n_features: 64
n_message_passing_blocks_cpainn_embed: 2
n_message_passing_blocks_cpainn_score: 5

N_max: 1000
length_scale: 3.
beta_scheduler: sigmoidal(-8,-4)
diffusion_steps: 1000

batch_size: 128
learning_rate: 1e-3
optimizer: Adam
```

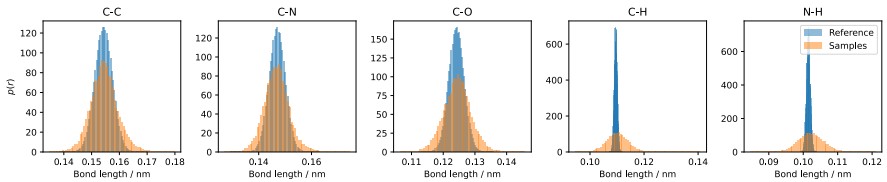

Figure 12: Bond lengths of samples Alanine Dipeptide

`n_message_passing_blocks_cpainn_{embed/score}` refers to the number of message passing and update blocks in the CPaiNN networks shown in Figure 2. A *message passing block* refers to a message block followed by an update block as shown in Figure 11. Where `sigmoidal(t_0,T)` $= \frac{1}{1+e^{-x}}\big|_{x\in(t_0,T)}$. `n_features` and `batch_size` corresponds to $n$ and $m$ in Figure 11. `length_scale` correspond to the value of $l_0$ in (Eq. 21) and defines the radial resolution of the embedding. `n_features` was chosen such that equivalent models could fit in memory of available hardware while maintaining a consistent `batch_size` across all systems. The remaining hyperparameters were fixed and were not systematically optimized.

### E.4 Bond lengths Alanine Dipeptide

We evaluate how well the fast vibrational degrees of freedom are captured by the SE3-ITO model on Alanine dipeptide by inspecting the bondlength distributions of model samples (Fig. 12). The variances are generally over estimated slightly, but it does not appear to significantly our ability to predict slow dynamics. However, it would impact importance sampling as many configurations would have unfavorable physical energies. We leave it for future work to improve.

## F  Compute resources

### F.1 Training

All reported experiments have been conducted on NVIDIA TITAN V, NVIDIA TITAN X (Pascal), and NVIDIA GeForce GTX TITAN X's. All GPUs have $\sim$ 12GB memory and range from 3000-5000 CUDA cores. Given the hyperparameters specified above, the SE3-ITO models converge within 2-4 days of training depending on system size.

Throughout the project, $\sim$ 250 models were trained for an average duration of $\sim$ 12 hours pr model on single GPU devices, resulting in a total of $\sim$ 3000 GPU hours spent on training.

### F.2 Sampling

In total 589 GPU hours have been spent on sampling throughout the entire project.

## G  Variational Approach to Markov Processes (VAMP)

The Variational Approach to Markov Processes (VAMP) is a recent result in non-linear dynamics theory, its key contribution is a family of VAMP-scores [21]. The VAMP-scores are devised based upon the insight that the best (smallest prediction error) linear model can be expressed in terms of the top singular components of the Koopman operator, $\mathcal{K}$ [79]. The scores measure sum of the singular values of $\mathcal{K}$ multiplied by overlap coefficients between a set of (ortho-normalized) feature-maps $f$ and $g$ and the singular components of $\mathcal{K}$. We can optimize VAMP-scores to learn optimal feature mappings and Markovian models of the dynamics from time-series data [65] or for model comparison [21]. We here use the VAMP-score for the latter and assume $f = g$.

VAMP-$r$ score is computed via the singular values of the Koopman matrix $\mathbf{K}$ estimated from data using the feature maps $f$ and $g$ [80],

$$\text{VAMP}-r = \sum_{i=0}^{k} \sigma_i^r \tag{22}$$

where $r \in \mathbb{N}_+$.

## G.1 VAMP gap

Informally, the VAMP-$r$ scores quantify the meta-stability of a Koopman matrix. We define the VAMP-gap $\Delta V$ between two Koopman matrices, $\mathbf{K}$ and $\mathbf{K}'$, as the difference between their VAMP-2 scores:

$$\Delta V = \text{VAMP}{-}2(\mathbf{K}) - \text{VAMP}{-}2(\mathbf{K}'), \tag{23}$$

where $\mathbf{K}'$ is a reference and $\mathbf{K}$ is a query matrix, respectively. In this context, $\Delta V = 0$ means meta-stability in $\mathbf{K}$ and $\mathbf{K}'$ is indistinguishable, $\Delta V < 0$ means $\mathbf{K}$ underestimates meta-stability, and *vice versa* for $\Delta V > 0$.