# OpenReview forum: "Implicit Transfer Operator Learning: Multiple Time-Resolution Models for Molecular Dynamics"
_NeurIPS.cc/2023/Conference — NeurIPS 2023 poster_

### Official Review · Reviewer_Ru8L · 2023-06-29

**Soundness:** 2 fair
**Presentation:** 3 good
**Contribution:** 2 fair
**Rating:** 4
**Confidence:** 2

**Summary:**

The authors propose Implicit Transfer Operator (ITO) learning, which aims to learn a surrogate of a molecular dynamics (MD) simulation process. Since standard MD simulations integrate Newton's equations of motion numerically, small integration time-steps are necessary, making simulations costly when processes on long time-scales (requiring many integration steps) are of interest. The proposed ITO method learns to simulate the dynamics across multiple time-scales, allowing to study processes on long time-scales more efficiently by using large time steps. It is also shown that ITO models are able to learn surrogate dynamics using partial observations, which is useful e.g. in the context of coarse-graining.

**Strengths:**

The presented method has strong theoretical foundations and the underlying theory is well-described.

**Weaknesses:**

While the premise of the paper is interesting, the results and contributions are not particularly impressive:
1. ChiroPaiNN is an extremely simple modification of the existing PaiNN architecture. So simple in fact, that I find it hard to argue this is a separate contribution (the only modification seems to be an added cross product between vector features, followed by a scalar product).
2. The authors write that their models show quantitative agreement with dynamic and stationary observables, which is not backed by the results shown in Table 2. Free energy differences of folding have relative errors in excess of 400% and also absolute errors are extremely large with over 3 kT in some cases. Mean first passage times have similarly large errors. These results cast doubt on the practical usefulness of the proposed method.

**Questions:**

- It is unclear to me why symmetry with respect to parity inversion needs to be broken for the task at hand. Can the authors explain why this is necessary? What happens when an E(3)-equivariant models is used instead of an SE(3)-equivariant one? Additional ablations studies in this direction would be insightful.

- Recent work (see arXiv:2302.00600) has pointed out an interesting connection between the score field $\nabla \log p$ (see Eq.7) in diffusion models and force-fields. Are the authors aware of this connection? Have the authors tried to "extract" the effective force-field learned by their model? My gut feeling is that it might be possible to run ordinary MD simulations with the learned force-field, but possibly using much larger time steps than would typically be feasible. I encourage the authors to investigate and run additional experiments in this direction.

- Can the authors think of ways to improve their method to achieve better quantitative agreement for observables like free energy differences with conventional MD?

- Direct efficiency comparisons to conventional MD would be interesting. What is the expected effective speed up achievable with the ITO method? Considering that training data for ITO needs to be generated, the models needs to be trained, and finally evaluated, I wonder how big the advantage is compared to directly running MD simulations.

**Limitations:**

Limitations with respect to transferability and scalability of the proposed method and the accuracy of the surrogate dynamics model are mentioned. In some cases it is unclear how the limitations can be reconciled in the future (e.g. the requirement of a closed-form expression for target path probabilities). Potential societal impacts of the work are not discussed, which however, I find acceptable considering the topic of the manuscript. A discussion of societal impacts would probably appear contrived.

---

> ### Author Rebuttal · Authors · 2023-08-09
>
> Thank you for taking the time to review our manuscript and providing insightful comments and questions. We believe your input will help us shape a much improved camera ready version. Please find commentary and replies to your concerns and questions below.
>
> _Comments on weaknesses:_
> 1 and 2. We believe that the modification, while indeed “simple” , is still a contribution as it solves a specific problem we had: we needed a highly efficient architecture with SE(3) equivariance (see clarification below). The values in Table 2 are generally very good considering there are no other methods which offer estimates of these values at all. Nevertheless, after submission, we found that training the models longer, led to better agreement between the MD and ITO values, suggesting that we systematically underfitting our models due to time-constraints. We now provide updated values for Chignolin and show a plot of convergence of observables as a function of training time in the global reply (Global reply fig 2 and table 2). We will update all values for the camera ready and provide convergence plots in the updated appendix.
>
> _Replies to questions:_
> 1. E(3)-equivariant models do not have the capacity to distinguish between chiral configurations. As a consequence, if we don’t use an SE(3)-equivariant score model, ie. symmetry is not broken when inverting the orientation, half of the predicted configurations will have the opposite chirality of the conditioning configuration. This is a structural problem because the model can not tell a configuration and its mirror image apart. In the Global response we illustrate this problem in Figs. 3 and 4. We will elaborate this discussion for the camera ready version.
> 2. This is indeed interesting work (two for one, Arts et al). We expect for large N that -\nabla_{x_N\tau} log p(x_N\tau|x_0) we will approach the effective force field since the density will converge to the Boltzmann distribution (see our appendix A.1). We have not tried to extract the forcefield from our current models, as learning a machine learned potential is not the focus of this work and we believe it would distract attention from the main contributions of this work. We do not believe that the learned forcefield would necessarily be amenable for simulation steps as large as those ITO enables, it is unclear what the reasoning here is.  Since our focus is on sampling and avoiding MD simulations, we leave these experiments for future work.
> 3.  As outlined above, we found that we had systematically under fitted our models due to time-constraints. After submission we found much better agreement, and show convergence of observables in our global response.
> 4. In the manuscript we do provide timings of sampling. The sampling speed in the current implementation declines quadratically in the number of particles for the simulation (appendix C.3, and limitation section) whereas MD scales from Nlog(N) to N^3 depending on different factors. Even so, with ITO we can stably simulate at a rate 6 orders of magnitude faster than MD. Nevertheless, current timings are not directly comparable, as MD is subject to other degrees of freedom which we ignore for now. To be specific, since the model is not transferable currently, we need MD data before we can do sampling for each system. In the strict sense, this means that in its current instance, there is no gain compared to MD, for an arbitrary new case. Nevertheless, we argue that by systematic data curation and parameter sharing, we can achieve a transferable ITO model in the near future which can capitalize on the multiple orders of magnitude speed ups presented. We transparently discuss this limitation in our manuscript and believe that in spite of it, the advances presented in this paper are extremely interesting for the broader AI4Science community.

---

> > ### Comment · Reviewer_Ru8L · 2023-08-13
> >
> > I thank the authors for their detailed reply and clarifications.
> >
> > 1. It is not true that E(3)-equivariant models do not have the capacity to distinguish between chiral configurations. For example, libraries like e3nn separate features into irreps of O(3), so they have "separate channels" for scalars (not able to distinguish mirror images) and pseudoscalars (able to distinguish mirror images). The proposed modification to the PaiNN architecture mixes pseudoscalars and scalars into one channel (breaking the symmetry w.r.t. to parity inversion). It is still unclear to me why that is necessary when pseudoscalar channels from an E(3)-equivariant model could also be used to distinguish mirror images, without needing to break symmetry.
> >
> > 2. Regarding my reasoning about extracting the learned force field from the model (point 2): I simply find this is an interesting test to consider, which could illuminate what the model has learned. As I explained in my original comment, the learned force field might allow much larger time steps than conventional MD, so there could still be a substantial speedup compared to conventional simulations. Of course, it is up to the authors whether they want to explore this direction or not.
> >
> > 3. I am glad to see you get improved results when training for a longer time!
> >
> > 4. I thank the authors for clarifying and I agree that a transferable ITO model that provides actual speedup over MD would be much more interesting than the current model.

---

> > > ### Author Response · Authors · 2023-08-13
> > >
> > > Thank you for engaging with our rebuttal. A few comments on your replies.
> > > 1. Including a pseudo-scalar to an E(3) equivariant network would make it SE(3) equivariant -- we achieve the same thing through a different means, as the reviewer outlines. We are not using e3nn, since it did not have the performance we needed for our application. We did use it initially, but had to leave it behind due to speed issues. We recognize e3nn is under heavy development and may in the future be suitable for applications like ITO.
> > >
> > > 2. If the reviewer could clarify their position, as to why and how they expect a Langevin type MD simulation in a machine learning potential would enable larger time-steps we would be very grateful. We have limited resources at our disposal and have to chose our experiments very carefully. We emphasize, that ITO does not rely on MD (integration of an SDE), and as such we are not sensitive to discretization errors which limit the time-step accessible in classical MD.
> > >
> > > 4) We are transparent about the limitations in the original manuscript, and yes we agree the model would be much more interesting if it was transferable at this point. However, keep in mind that this is a big ask, currently no ML coarse-grained models are transferable, and no existing methods allow for long-time scale simulations like ITO.

---

> > > > ### Comment · Reviewer_Ru8L · 2023-08-13
> > > >
> > > > 1. It seems the authors are not fully aware what SE(3)-equivariant/E(3)-equivariant mean, respectively. SE(3) is the special Euclidean group, consisting of rotations and translations (and combinations thereof). E(3) is the Euclidean group, consisting of rotations, translations, and reflections (and combinations thereof). To be equivariant w.r.t. to a group means, loosely speaking, that it does not matter whether a group action is applied to the input or output of a function, the result is the same. An E(3)-equivariant architecture can distinguish between pseudoscalars and scalars just fine, that does not make it SE(3)-equivariant. If something is SE(3)-equivariant (instead of E(3)-equivariant), it means that this "input-output symmetry" w.r.t. to reflections of the coordinate system "is broken". This is exactly what happens in your network: Since pseudoscalar and scalar components are mixed together, a reflection changes the outputs in a manner that is not well-defined, because scalar components do not change sign, while pseudoscalar components do. Regarding e3nn: I never wanted to suggest the authors should use e3nn, I mainly gave it as an example (since it is well-known), so the authors could convince themselves that what I am saying is correct. I will not engage in further discussion about the meaning of SE(3)/E(3)-equivariance - these are well-defined concepts with a clear meaning and I can only encourage the authors to check that what I am saying is correct. Why it is necessary to break symmetry is still not clear to me, using pseudoscalar components of an E(3)-equivariant setup would be sufficient to distinguish mirror images.
> > > >
> > > > 2. I merely suggested this experiment as an interesting test, whether the authors want to perform it or not is up to them. Usually, the time step during MD simulations needs to be small because of high-frequency components in the dynamics. It might be that the "effective force field" their model has learned "damps" high-frequency motion and instead gives only low frequency dynamics. I personally would find this very interesting (I can only speak for myself).
> > > >
> > > > 4. I think there is no controversy here, we all agree that a transferable model would be much more interesting. I never wanted to suggest that achieving this would be an easy task - on the contrary. Hence, the method would be much more interesting to the community if it was transferable, because as the authors correctly state, other methods that are not already exist.

---

> > > > > ### Author Response · Authors · 2023-08-14
> > > > >
> > > > > 1. Apologies for the mix up. The reviewer is indeed right about our mistake. We did a patch solution to PaiNN to make it not sample mirror images with equal probability as we present in our figures 3 and 4 in the global response. We will revise the text accordingly and investigate proper E(3) equivariant architectures in the future.
> > > > > 2. Interesting point. These things are indeed investigated in the coarse-graining literature, but is not clear whether learning a sampler together with a potential will make the potential smoother than regular potential estimation would.
> > > > > 3. The reviewer is trying to make our paper about something which we never claim and in fact state as a limitation. We will not engage in further discussion on this topic.

---

### Official Review · Reviewer_r4Zz · 2023-07-01

**Soundness:** 4 excellent
**Presentation:** 1 poor
**Contribution:** 2 fair
**Rating:** 6
**Confidence:** 4

**Summary:**

The paper develops a conditional diffusion model to sample unnormalized densities - e.g., the Boltzmann distribution of molecules to replace molecular dynamics simulations. Given a molecule structure, noise is added to it and the diffusion model denoises this distribution to the distribution generated by MD. The diffusion model is not only conditioned on diffusion time, but also on MD time, such that a single model can be used to make larger or smaller MD steps.

The method is not generalizable - training data needs to be collected via the expensive MD that the method aims to replace. The paper is transparent about this. Experiments are performed on a toy system, alanine dipeptide, and 5 fast folding proteins.

The paper has strong contributions but is very unpolished. I think it can be substantially improved in the rebuttal, especially by addressing my first two content concerns in the weaknesses.

I thank the authors for any time they take to answer my questions!

**Strengths:**

### Strengths

1. The authors tackle an important and impactful task (sampling Boltzmann distributions of molecules).
2. The idea is clever. Since it is hard to sample the Boltzmann distribution of a molecule unconditionally, the authors instead construct a noisy distribution that still shares structure with the bolztmann distribution such that the generative model has an easier job at producing a sample from the less complicated conditional distribution instead of the unconditional one. **************************************************Notably this was already done by TimeWarp************************************************** but the authors bring diffusion models to this task which seem better fit for it (if no reweighting is done).
3. It is an interesting hypothesis that training with multiple MD times improves stability and would perform better than a single MD timestep size. Furthermore, this is justified and illustrated with interesting transfer operator theory. I think there is great value in bringing this theory to the attention of the ML for MD and boltzmann generator community.
4. The model is self consistent. The experiments set up for this are ideal and illustrate the self consistency very well.

**Weaknesses:**

The first three content concerns are by far the most relevant concerns

### Content concerns (approximately in order of importance)

1. The main claim (as I see it), that training with multiple MD times helps, is not sufficiently investigated:
    1. Why do you only show results for the Muller Potential for this? I do not see why not to evaluate it for all systems or at least for some molecule instead of only the toy system.
    2. Why do you only show VAMP2 score gaps and not kl divergences?
    3. What about 1000 lag in Table 1?
2. What is the comparison in wallclock time for sampling the whole distribution compared to MD? I would expect to see e.g., a plot of how the KL divergence of some ovservable goes down in MD vs. with your method in wall-clock time. Running diffusion models also takes quite some time I imagine, so is this not one of the main things that need to be investigated? Am I missing this information somewhere in the paper?
3. The paper claims to be reproducible, but there is no code provided.
4. Is there related work on ML for transfer operators (e.g., by Anima Anandkumar) that should be explained in the related work? I am not familiar with the field - just a potential pointer.


### Presentation Concerns

I think the presentation has to be improved and currently suffers from missing information, information relegated to the appendix, distraction with irrelevant material, and unnecessarily complicated explanations. The first two concerns are the most relevant.

1. Are we obfuscating the method unnecessarily with math to make it seem more sophisticated? I think it would be tremendously helpful if the transfer operator theory is explained and justifies correctness, but the explanation of what is actually done in the end can be vastly simplified and made much more transparent by just stating that it is now a diffusion model that is conditioned on the MD timestep as well and can therefore make larger or smaller MD timesteps during inference.
2. The “novel” architecture explanation is unnecessary and it is really not needed to claim this as a contribution instead of just focusing on the clearly good contributions and the point of the paper (to me at least) in using multiple MD times and a conditional diffusion model to replace MD. These are two novel **************actual************** contributions. How about just saying you use Painn and there is a little tweak similar to what you can do with e.g. e3nn to deal with chirality that the reader can look up in the appendix?
3. (would take quite some rewriting I suppose): I think the presentation might benefit a lot from distinguishing between e.g., an “MD SDE” and a “Diffusion SDE”. Then you can distinguish between an MD time and a diffusion time and say that in the text next to your superscript, subscript notation.
4. Figure 3 does not show any visible difference between fixed and stochastic lag? My recommendation would be to put it in the appendix and use the space for something useful.
5. I would recommend not talking about a special coarse grained approach in the contributions introduction and abstract. I went into the paper thinking you do something interesting with coarse graining.
6. You talk about nested sampling vs. direct sampling. While it is clear to me what is meant, this is not explained anywhere. A simple sentence would suffice.
7. More explicitly point out the dependence of the added noise on the size of the MD timestep size.
8. It should be mentioned in the main text that the number if diffusion ODE steps is the same independent of the size of N.
9. Maybe fix all the typos especially in the related works section for the next/camera-ready version :sweatsmile:
10. The appendix pointer for Algorithm 1 is wrong.

**Questions:**

### Questions

1. For the dependence of the diffusion/the amount of added noise on the size N: What is the connection between increasing the maximum diffusion time T and making beta_i dpend on i instead and controlling the variance with that? Is one correct and the other is not or are they equivalent?
2. Did you try an SDE solver for the diffusion SDE?
3. How are the standard deviations in Table 2 calculated?
4. Why do you use the same number of diffusion steps/discretization steps if you have a higher or lower N?
5. How many samples do you draw in all your experiments? How long do you run the “MD” with the diffusion model?
6. Why does the same i index both t and N? Wouldnt that mean that a particular step in the trajectory always uses the same N?

**Limitations:**

The paper is transparent with the main limitation, that there is currently no capability to generalize. I think the extent to which this is made clear could go further though: point out that the model is only **********************overfitting********************** on a distribution and that there is **********no "real-world use value" (e.g., speeding up MD) in its current form.**********

The paper also points out the important difference to e.g. Timewarp that there is no direct way to do reweighting and get exact convergence to the boltzmann distribution in the limit. However, in the discussion it is made to seem like obtaining exact likelihoods from a diffusion model is very much feasible as well and would not come with a large amount of difficulty.

---

> ### Author Rebuttal · Authors · 2023-08-09
>
> We thank the reviewer for taking time to read our manuscript thoroughly, and providing their insightful comments. We in particular appreciate the comments about the presentation, which will help us prepare a much improved camera ready version. We are addressing their highlighted weaknesses and questions in a point-by-point reply below.
>
> _Comments on weaknesses:_
> 1. We show for MB as it is the system which can be extensively sampled using conventional methods and where statistics can be computed with confidence. We have trained models for Alanine dipeptide with fixed lags and include VAMP2 gaps calculated for these models as well for completeness, preliminary figures are attached in the Global response (Fig 1), and ecco the results reported for MB. Final figures will be included in the camera ready version. KL divergences are insensitive to low density regions per construction, capturing slow dynamics means capturing low density regions well. VAMP2-gaps compare the singular values of a Koopman operator and directly measures meta-stability, e.g. how well we capture the gaps between high and low density regions of the configuration space. We used trajectories of length 1000\tau to generate the statistics in the table, to ensure the same statistics across the lag times. The estimation of the Koopman operator is not stable when the lag is equal to the length of the trajectory, so we cannot compute the VAMP2-gap reliably at 1000\tau. We will redo these calculations for the camera ready versions and include all lags.
> 2.  We provide estimates of wall-clock time per conditional sample for the systems in the appendix C.2. We also discuss this briefly in section 4.3 in the main text for one example (Chignolin). The main difference is that for MD, we need to run simulations for N\tau steps to draw one sample for the Monte carlo estimator (2. Dynamic observables). With ITO we can draw one sample at N\tau in a 1-shot fashion, and for coarse-grained Chignolin drawing one sample takes ~4 ms. If we very conservatively say we can simulate a coarse-grained Chignolin at 100 microseconds per day with similar GPU resources, and want to evaluate a correlation function at N\tau = 1 microsecond, it means we can get 100 independent samples per day for a TitanX, for ITO we would get on the order of 10^7 independent samples. We realize that this discussion has been lackluster in the manuscript and will improve it for the camera ready version.
> 3. We will provide code when the paper is accepted for publication.
> 4.  We have been in active conversation with Prof. Anandkumar and solicited feedback from her group about ITO, and while we got pointers to work on ODEs and PDEs using their Neural Operators approaches, they did not so far investigate transfer operators to our knowledge. If the reviewer can point us to specific references we are happy to expand this discussion.
>
> _Replies to questions:_
> 1.  Interesting idea, we have not explored this. We encode the physical time (N) using a positional encoding, which then defines the conditional diffusion model, and in turn the distribution of the random variable x_N\tau | x_0.
> 2.  No. However, we are using ODE solvers to solve the probability flow ODE, for efficient inference of the trained model. We discuss this in the appendix and provide timings for inference.
> 3.  The uncertainties are estimated using Bayesian posterior sampling of Markov models consistent with the data (using procedure described in Trendelkamp-Schroer and Noe JCP 2013). We will clarify this in the camera ready version.
> 4.  We have not experimented with varying the number of diffusion steps for different N. However, the motivation to keep it the same is that the conditional density at different N are combinations of the same functions but weighed differently (see. Eq. 8, and Appendix A.1.). Since we do not know the weights a priori, we delegate this to a positional encoding. The hope was – and what we see is – that ITO is indeed able to generalize across time horizons (N), even if we do not represent the operator explicitly.
> 5. Each plot in Figure 4 is calculated using 15,000 trajectories. For direct sampling, this means that we had to take 15,000 samples in total. In the case of ancestral sampling, we had to sample 4/64/512 steps for 15,000 trajectories, depending on the value of \Delta t.
> The contour lines of the fast folding proteins in Figure 5-8 were calculated using 10,000 trajectories. The density in the background are samples from the reference trajectories. The simulation step is 200ps and the simulations were performed with ancestral sampling. This means that in order to plot the contour lines of the \Delta t = 200ns we had to perform 1000 steps. We will clarify these details for the camera ready version of the manuscript.
> 6. The i index in the context of the mini-batch training is indicates a tuple of B_i=(x_{t_i}, x_{t_i+N_i\tau}, N_i), where the t_i and N_i are sampled randomly for each batch, i. The i is an index in the batch, which is composed of conditioning state, time-lagged state and time-lag (horizon). We realize this notation is not standard and might be confusing. We are revising it for the camera ready version.

---

> > ### Comment · Reviewer_r4Zz · 2023-08-13
> >
> > Thank you for the detailed responses and thoughtful explanations.
> >
> > 1. Thank you for the VAMP2 clarification. Figure 1 in the new pdf: the average over this scattered performances over epochs, with the average being worse for lag 100, is not very convincing for the argument that training with multiple lags will likely give me a better model than a fixed lag. What if we had epochs 50 to 60 - is 40 to 50 cherry picked?
> > 2. Is my understanding incorrect that this does still not provide the same information as, e.g., a KL divergence decrease over time would provide? If the samples of ITO are bad, it will not matter how fast they are generated - the KL divergence (or better metrics) will not decrease.
> > 3. If you do not provide code, I think it is incorrect to claim that the results are reproducible.
> > 4. I should not have listed this as a concern, it is a mere question whether neural operator learning is relevant here and I cannot provide specific points that should be discussed in the references.
> >
> > The time taken to answer my questions is highly appreciated.

---

> > > ### Author Response · Authors · 2023-08-14
> > >
> > > Thanks for engaging in the discussion!
> > > 1. No these are not cherry picked, they are the last 10 epochs of training, where the loss has stabilized. The fluctuations are large, but the averages are converged. In all cases but lag 100 ITO is better, and in that case ITO and fixed lag are statistically indistinguishable. We would say that this is a clear out performance. Consider the ITO model is a single model predicting all time-scales, and fixed time-scales require a new model for every time resolution to be estimated.
> > > 2. Thanks for the clarification. Making such a comparison is challenging for any of the molecular systems due to the computational requirements incurred. But we will include such a plot for MB in the appendix for the camera ready version. Thanks for the suggestion.
> > > 3. We agree, and consequently we will release the code once the paper is published to comply with the NeurIPS recommendations.

---

> > > > ### Comment · Reviewer_r4Zz · 2023-08-15
> > > > **Convinced.**
> > > >
> > > > Thank you for the further details!
> > > >
> > > > 1. I think this is then indeed good evidence that training with multiple timescales helps.
> > > > 2. I was amiss and now think lesser of the relevance of this experiment. It is quite clear that ITO is by far faster.
> > > >
> > > > How are you planning to change the presentation of your paper?
> > > >
> > > > I now also realized that my assessment "but the authors bring diffusion models to this task which seem better fit for it (if no reweighting is done)." was wrong and that this was previously done.
> > > >
> > > > Is it correct that your main claim now is that using multiple time scales helps? Are planning to put this as the main contribution of your paper?

---

> > > > > ### Author Response · Authors · 2023-08-15
> > > > > **Thank you!**
> > > > >
> > > > > We appreciate the reviewers honesty!
> > > > >
> > > > > We plan to make the following changes in the presentation:
> > > > > - emphasize the multiple time-scale training in the main contributions
> > > > > - move ChiroPaiNN architecture away from the main contributions, and mention it as a minor contribution in results section.
> > > > > - reorganize Section 3 to first present what is done, and then provide a justification to make easier for readers to identify what was precisely done.
> > > > > - Figure 3 will be moved to the appendix, and the space made available will be used to expand and clarify the discussion of the ITO and the theory, specifically several of the presentation concerns highlighted by the reviewer in the original manuscript.
> > > > > - we will add a VAMP-2 score plot and refined versions of the plots in the global response to the appendix of the manuscript and expand discussion surrounding these.
> > > > > - We will make the changes highlighted in our rebuttal to this reviewer and the other reviewers.

---

> > > > > > ### Comment · Reviewer_r4Zz · 2023-08-15
> > > > > > **Raising score. Suggestion.**
> > > > > >
> > > > > > Thank you for the time taken to clarify, to answer my questions, and for the informative discussion.
> > > > > >
> > > > > > I think the new results make this an interesting contribution, although I realized one of my initial strengths was wrong. I recommend acceptance and raise my score.
> > > > > >
> > > > > > In my opinion, it would also be a positive change to already clarify in the introduction and to more explicitly make the point than it is currently the case in the limitations section, that the model is only overfitting on a distribution and that there is no "real-world use value" (e.g., speeding up MD) in its current form.

---

> > > > > > > ### Author Response · Authors · 2023-08-16
> > > > > > >
> > > > > > > We thank the reviewer for their suggestions and for raising their score. We will clarify the current limitation of ITO in the introduction in the camera ready version.

---

### Official Review · Reviewer_Dude · 2023-07-03

**Soundness:** 3 good
**Presentation:** 2 fair
**Contribution:** 2 fair
**Rating:** 5
**Confidence:** 2

**Summary:**

This paper presents implicit transfer operator (ITO) learning for the simulation in molecular dynamics. Their approach adopts the SE3 equivariant MPNN architecture (ChiroPaiNN) to parameterize the transition kernels in the denoising diffusion probabilistic model. The method displays a decent performance on several all-atom molecular simulation tasks with only coarse-grained representations.

**Strengths:**

- The problem is very relevant and the motivation is clear. Most notably, the method takes SE3-equivariance into consideration, which I think is a very good practice to combine state-of-the-art generative models with the simulation of molecular dynamics.
- The paper presents a good balance between the introduction, methodology, and experiments. It is an interesting paper to read.
- Illustrative examples are shown to demonstrate the effectiveness of their approach with detailed implementation details.

**Weaknesses:**

- The theory part concerning transfer operators is chaotic. Eq. (3) has a typo in it ($\rho$ should be $f$). And in Appendix A.2, I don't think Eq. (17) should be correct if the transfer operator is defined as in (16). It seems they mix up the definition in Eq. (16) with that defined in Eq. (5) of [1].
- The experiments seem sound but it is very hard for me to evaluate the effectiveness of their approach since there is a lack of comparisons. As listed in the related works section by the authors, the idea of transfer operator surrogates has already been commonly used in molecular modeling as well as deep generative Markov state models. I believe there would be state-of-the-art methods other than all-atom MD simulations, to which they are comparing.
- As I mentioned before, the use of an SE3 equivariant message-passing neural network seems one of the main contributions of this work. Following the last point, I would suggest at least one experiment that could explicitly demonstrate how and why this architecture is beneficial to the diffusion model. Will the results be much worsened if we remove this SE3-equivariance from the architecture, given a similar size of the NN?

[1] Prinz, Jan-Hendrik, et al. "Markov models of molecular kinetics: Generation and validation." The Journal of chemical physics 134.17 (2011).

**Questions:**

- I don't see a direct connection between Eq. (4) and (5) and their method. Would you explain more about the motivation of the algorithmic design from this "relaxation"?
- Following the discussion in Appendix A.1, it seems that when $N$ is very large, the spectrum of the transfer operator $T_\Omega$ will become very ill-posed (with one eigenvalue being 1 and all others converging to 0). Will this affect the performance of the method? Or does this mean that the $N_{\rm max}$ in Algorithm 1 should not be too large?
- I don't quite understand the results in Figure 5. There are two dashed lines in the figure all labeled "Folded".
- Following the previous point, there seems to be a large discrepancy between the value of $\langle \tau_f \rangle$ for Trp-Cage, BBA, and Villin in Table 2. Would you explain how should we interpret this? Is the MD result generally accurate or it is also just a reference? If so, how could we know that the simulation yields reasonable results?

---

> ### Author Rebuttal · Authors · 2023-08-09
>
> We thank the reviewer for their thorough review of our manuscript and the comments, in particular regarding the presentation of the maths and comparison to previous work. We believe the input will help us prepare a much improved version for the camera ready version.
>
> _Comments on weaknesses:_
>
> 1. There was indeed a typo in eq. 3 \rho and f were mixed up in that equation, we will fix this in the camera ready version. Great catch! The reviewer here refers to the propagator which is mathematically different from the transfer operator, but missed the scaling by \mu which makes the difference. Our eq. 17 and we refer to A.1, for a more verbose derivation in bra-ket notation.
> 2. There are no other methods available which allow for sampling on these time-scales and allow an apples to apples comparison. DeepGenMSMs are essentially deep Markov models, which encode configurations as state memberships and then model transition densities between discrete states in a latent space, with a Koopman operator. As shown in their original paper, these methods introduce severe distortions in the local structures when long time-scales are queried. So while DeepGenMSMs have realistic latent space dynamics, the real space dynamics cannot be faithfully reconstructed.  ITO does not suffer from this drawback.  The only methods which allow for sampling of long time-scales faithfully are conventional MD simulations and timewarp (arxiv:2302.01170); the latter paper is in preprint with no code or data available publicly, and suffer other severe drawbacks. Nevertheless, we recognize that expanding the section on prior work will help clarify the ITO contributions to the readership better.
> 3. We provide a figure in the global reply showing how a E(3) equivariant model samples both mirror images of alanine dipeptide with equal probabilities, whereas only one mirror image is accessible in the physical dynamics used to train the model. Introducing SE(3) equivariance overcomes this problem.
>
> _Replies to questions:_
>
> 1. Indeed, we do not directly use eq 4, 5 and 8, as they would involve optimization on the Stiefel manifold, and would require us to choose the number of eigenfunctions to estimate. We therefore model these expressions implicitly, by modeling the transition density. We use these equations to motivate the connection between the time-horizon N\tau and transition probability between two configurations x and y. We conjecture that we can leverage this underlying mathematical structure to instead train a single model which models the transition density at multiple different time horizons [1, N_max], since the proportions of different functions vary exponentially in N. Indeed, we see that training a model with multiple N is an advantage: we get better prediction of meta-stability when training with multiple time horizons.
> 2. No, we have not experienced any negative impact of large time-scales in our experiments. Since we converge to the Boltzmann distribution for large N, e.g. the state x_N\tau ~ \mu(x) independently of x_0, this is the hallmark of ill posedness, but it is also what we expect from the dynamics: at sufficiently long time-scales we expect our Markov chain to mix instantaneously.
> 3. We are illustrating two collective variables tIC1 and tIC2 in our analyses. The dashed lines correspond to the tiC1 and tiC2 coordinates of the folded state. For clarity, we will change the style of the dash to distinguish the two tICs for the camera-ready version.
> 4. We expect these values to align within an order of magnitude. However, after submission, we found that training the models longer, led to better agreement between the MD and ITO values. We now provide updated values and show a plot of convergence of observables as a function of training time in the global reply for Chignolin. We will include these updated values for all four proteins in the camera ready version.

---

> > ### Comment · Reviewer_Dude · 2023-08-20
> >
> > Thank you for the detailed response. I am raising my score to (5) but with less confidence (2).

---

> > > ### Author Response · Authors · 2023-08-20
> > >
> > > Thank you!

---

### Official Review · Reviewer_BcfP · 2023-07-09

**Soundness:** 3 good
**Presentation:** 3 good
**Contribution:** 3 good
**Rating:** 7
**Confidence:** 2

**Summary:**

The proposed approach aims to learn molecular dynamics via an implicit transfer operator framework that can perform modeling at multiple time scales. The framework is using diffusion model with SE3 equivariant architecture. The approach has shown capability in stable and self-consistent modeling at multiple time and space resolutions for molecular dynamic modeling.

**Strengths:**

- a novel framework for multi-scale simulation for molecular dynamics, combined with the SE3 network for stable, long-term results.
- demonstrate good experimental results on multiple MD datasets and achieve an order of magnitude speedup
- work on an important task of molecular dynamic modeling
- use diffusion network for exact likelihood evaluation
- honest and clear notes on limitations - appreciated

**Weaknesses:**

- in terms of presentation, it'd be nice to make each figure self-contained in terms of points to make and the meaning of the notations.
-

**Questions:**

- how are the number of time scales chosen during the training?
- could you explain in the caption or here all the input for ITO networks in Figure 2
- what's the difference between the diffused time and actual time

**Limitations:**

- this paper may be intense for people noting working on MD. it'd be nice to include more descriptions in captions.
- I am curious how it can be generalized to other dynamics like some other physics system that is also multiscale.

---

> ### Author Rebuttal · Authors · 2023-08-09
>
> We would like to thank the reviewer for their favorable and supportive evaluation and helpful comments and questions on our paper. We reply to the questions below and welcome a discussion about any outstanding doubts.
>
> _Questions:_
>
> 1. The number of time-scales chosen during training, can be understood as how many linear combinations of the eigenfunctions we want the model to see during training (eq.8). We set this number (N_max) to 1000 initially, and we did not experiment with varying this value. We leave optimization of this hyper parameter to future work.
>
> 2. z is an atom-embedding (nominal embedding), N is the multiple of the physical time-step \tau (positional embedding), x_0 is the cartesian coordinates of the conditioning state, t_diff is the diffusion time, and x_{0+N\tau} is the partially denoised state of the time-lagged configuration given x_0. Note that there is a hat missing on the   x_{0+N\tau} to be consistent with the main text, we will fix this for the camera ready version. B) Variable names have the same interpretation, and there is also a hat missing on x_{0+N\tau}.
>
> 3.  The diffusion time is the progress along the diffusion process which models the conditional probability density p(x_N\tau | x_0), the physical time (“actual time”) is the time-step N\tau, and is connected to the physical process which we are modeling.

---

### Official Review · Reviewer_nRNz · 2023-07-22

**Soundness:** 2 fair
**Presentation:** 2 fair
**Contribution:** 3 good
**Rating:** 5
**Confidence:** 4

**Summary:**

In this work, the authors proposed a framework that combines SE(3)-equivariant MPNN and conditional DDPM, called Implicit Transfer Operator (ITO) learning, as a method to efficiently sample observales from MD simulation trajectories. Such a framework is validated on Muller-Brown potential data generated by the authors, as well as the commonly used alanine dipeptide and fast-folding protein dataset generated using MD simulation. ITO learning framework is shown to bear the potential of accelerating or surrogate MD simulation.

**Strengths:**

1. It is a nice effort to formulate the MD trajectory using the ITO such that the sampling can be learned by the conditional DDPM.

2. Proposed a SE(3) version of the PaiNN.

**Weaknesses:**

#### Major
My major concerns are related to the experiment and results sections of the manuscript.
1. In Figure 4, the $\phi$-$\psi$ plot or Ramachandran plot is often colored by the free energy of the system (e.g. [Köhler et al. 2023](https://pubs.acs.org/doi/epdf/10.1021/acs.jctc.3c00016), [Marloes et al. 2023](https://arxiv.org/abs/2302.00600) and many more), which is directly related to the probability of the state given Boltzmann distribution. From the current Figure 4, the physical property of the system, energy, is not directly visualized.

2. Another issue with Figure 4 is the large bin size of the 2D histogram. The current binsize of 2D histogram is about 0.4 rad, which makes it really hard for readers to understnad the performance of the SE3-ITO model. Moreover, the MD simulation data are represented in the form of 2D histogram while the model sampled data are in the form of contours. I would highly recommend that the authors to show the MD data in separate subfigures with the same form of the model sampled data. If 2D histogram is to be used, a much smaller bin size should be used for clarity.

3. Although the effort of proposing a SE(3)-equivariant version of PaiNN (ChiroPaiNN) should be recognized, the necessity of CPaiNN is not well estabilished in the manuscript. As the author mentioned in the manuscript, there is no parity change during MD simulation. I am wondering if there will be significant difference in accuracy if the original PaiNN is used in the ITO learning framework. Assuming an SE(3)-equivariant model is absolutely necessary, the authors have not shown any comparison between CPaiNN and other established SE(3)-equivariant model such as the [SE(3)-Transformer](https://arxiv.org/abs/2006.10503). Such a benchmark can definitely help to improve the manuscript.

4. For the fast-folding protein experiments, CG-SE3-ITO model is compared with MD data. In Table 2, the error of the model is significantly higher for proteins with more residues or $C_{\alpha}$ atoms. Yet, the discussion about the high error is limited in the manuscript. The purpose of coarse graining is to achieve relatively high accuracy with large system. Underwhelming accuracy on large system impairs the applicability of the SE3-ITO model on coarse graining.


#### Minor
5. Line 65, the probability is written as $p_{\tau} (x_{t+\tau} | x_{t})$, which is inconsistent with the notation in the dynamics observables equation ($p_{\tau} (x_{t+\Delta t} | x_{t})$) in line 64.

**Questions:**

1. In Figure 3, I assume that the green curve ($N=1000$) is not sampled from nested conditional probability. Can you verify my understanding?

2. In Figure 3, the model with ($N=10$ and $\Delta t=1000\tau$) is lower comparing to higher $N$ values. In Figure 4, $\Delta t=4$ps seems to lead to less accurate samples from model. My interpretation of those results is that the ITO learning framework suffers higher error when predicting more immediate conformational change of the molecule in MD simulation. If so, the dynamics of the molecule when moving between high probability state might be missed when using ITO framework. Can the authors discuss more about this?

**Limitations:**

Yes, limitations are addressed.

---

> ### Author Rebuttal · Authors · 2023-08-09
>
> Thank you so much for taking the time to carefully read our manuscript and providing constructive input and criticism as to how we can improve it. Below, you will find a point-by-point reply to your concerns and questions. We are looking forward to continue the discussion, please do not hesitate to ask if things remain unclear.
>
> *Comments to major weaknesses*
>
> Points 1-2: To clarify, Fig 4, does not show Boltzmann distributions but transition densities. Specifically, we show the log transition probabilities at three given time-lags (or time-horizons) from MD (coarse 2D and 1D histograms). SE3-ITO samples are overlaid with contour lines. As we are amongst the first to present work on models able to do this, we have experimented with visualization to make it as informative as possible. To clarify, the low resolution on the MD histograms are due to the high dimensionality of the transition density; it scales quadratically in the number of bins. Recall that we need to compute the probability from any bin to any other bin. For these 16x16 histograms we divide the data into 65536 transition counts, with 750000 MD samples in our training data we doubling the resolution would give us less than one sample per bin on average. In the plots we show only the transition statistics from the bin to which the initial condition is assigned.
>
> We would like to emphasize that, we do also compare SE3-ITO samples (blue and orange) and MD samples (black) in the marginal histograms.
> Nevertheless, we recognize the reviewers' concern about the comparison of statistics shown with different resolutions, and will include a 1- and 2-dimensional histogram of the model and training samples for more direct comparison, for the camera-ready version. 2D histograms will be separated into individual plots.
>
> Point 3: In the Global response we show why an SE(3) equivariant model is necessary for the molecular applications pursued here (figs 3-4). Briefly, if we use a parity invariant model, we will sample mirror images of molecules with equal probability a priori, even if one of the mirror images is inaccessible in the physical dynamics used to simulate the system. This is also a shortcoming observed in timewarp (arxiv:2302.01170), that uses a permutation equivariant architecture. While it is interesting to investigate and compare different SE(3) equivariant architectures for SE3-ITO models, we are confident that these are best addressed in future work, as such a benchmark would add little scientific value to the current manuscript
>
> Point 4: After submission we found our models were slightly underfitted, and after training longer we found improved agreements with observables across all proteins. We are providing new values in the global response table 2, for Chignolin, along with convergence plots. We will update all values for the camera ready version.
>
>
> _Questions:_
> 1. Yes this is correct. It is sampled directly with \Delta t = 1000\tau.
> 2. We have to be careful comparing N across different data-sets, with different resolutions in space and time. In general, we cannot make a statement as to whether fast (‘immediate’) dynamics are captured poorly by our ITO models. Our preliminary investigations of this (Fig 10, in the supplement) suggest that we indeed do not capture very fast dynamics perfectly. However, we argue that these fast dynamics do not need a model like ITO. We can already study fast dynamics very well with conventional MD simulations. Their limitation is the mixing between modes interconnected by low probability barriers. On that task ITO does extremely well.

---

> > ### Comment · Reviewer_nRNz · 2023-08-15
> >
> > Thank you for addressing my concerns and answering my questions regarding the manuscript. For the transition density plot (Fig.4), I gain deeper understanding of your transition density visualization after reading your explanation. I would highly recommend that the authors add more detailed explanation about Figure.4 to the camera-ready version of manuscript. Also, the Fig.3 and Fig.4 is sufficient to show the necessity of SE(3) equivariant model in this case. I suggest to briefly mention that in the manuscript.

---

> > > ### Author Response · Authors · 2023-08-16
> > > **Thank you**
> > >
> > > We are glad to hear that our more in depth explanation of our figures helped the reviewer understand the presented results better. For the camera ready version we will explain what is shown Fig 4 in more detail. We will tone down the discussion of the ChiroPaiNN architecture, but include the figures 3-4 (global response) to illustrate the need for the modification we introduce in PaiNN in the appendix.

---

> > > > ### Comment · Reviewer_nRNz · 2023-08-20
> > > >
> > > > Great! I have raised my score because of the revision.

---

> > > > > ### Author Response · Authors · 2023-08-20
> > > > >
> > > > > Thank you!

---

### Author Rebuttal · Authors · 2023-08-09

Global response. see attached pdf.

---

### Decision · Program_Chairs · 2023-09-21

**Decision:**

Accept (poster)

**Comment:**

The reviewers raised many concerns about the presentation of the paper. They all agree that the paper has some value but also agree that the paper, at its present shape, is not ready for submission.


We encourage the authors to consider the feedback provided in the review process.